



# Modeling the impacts of diffuse light fraction on photosynthesis in ORCHIDEE (v5453) land surface model

Yuan ZHANG[1, 2], Ana BASTOS[3, *], Fabienne MAIGNAN[2, *], Daniel GOLL[4], Olivier BOUCHER[5], Laurent LI[1], Alessandro CESCATTI[6], Nicolas VUICHARD[2], Xiuzhi CHEN[7], Christof AMMANN[8], Altaf ARAIN[9], T. Andrew BLACK[10], Bogdan CHOJNICKI[11], Tomomichi KATO[12,13], Ivan MAMMARELLA[14], Leonardo MONTAGNANI[15,16], Olivier ROUPSARD[17,18,19], Maria J SANZ[20,21], Lukas SIEBICKE[22], Marek URBANIAK[11], Francesco Primo VACCARI[23], Georg WOHLFAHRT[24], Will WOODGATE[25,26], Philippe CIAIS[2]

[1]Laboratoire de Météorologie Dynamique, IPSL, Sorbonne Université/CNRS, Paris, France
[2]Laboratoire des Sciences du Climat et de l'Environnement (LSCE), IPSL, CEA/CNRS/UVSQ, Gif sur Yvette, France
[3]Department of Geography, Ludwig–Maximilian University of Munich, Munich, Germany
[4]Department of Geography, University of Augsburg, Augsburg, Germany
[5]Institut Pierre–Simon Laplace, CNRS/Sorbonne Université, Paris, France
[6]Institute for Environment and Sustainability, Joint Research Centre, European Commission, Ispra, Italy
[7]Guangdong Province Key Laboratory for Climate Change and Natural Disaster Studies, School of Atmospheric Sciences, Sun Yat-sen University, Guangzhou 510275, China
[8]Climate and Agriculture Group, Agroscope, Zürich, 8046, Switzerland
[9]School of Geography and Earth Sciences and McMaster Centre for Climate Change, McMaster University, Hamilton, ON, Canada
[10]Faculty of Land and Food Systems, University of British Columbia, Vancouver, BC, Canada
[11]Poznan University of Life Sciences. Piatkowska 94, 60-649 Poznan, Poland
[12]Research Faculty of Agriculture, Hokkaido University, Sapporo 060-8589, Japan
[13]Global Institution for Collaborative Research and Education, Hokkaido University, Sapporo 060-8589, Japan
[14]Institute for Atmospheric and Earth System Research/Physics, Faculty of Sciences, University of Helsinki, Finland
[15]Autonomous Province of Bolzano, Forest Services, Via Brennero 6, Bolzano 39100, Italy
[16]Faculty of Science and Technology, Free University of Bolzano, Piazza Università 5, Bolzano 39100, Italy
[17]CIRAD, UMR Eco&Sols, BP1386, CP18524, Dakar, Senegal
[18]Eco&Sols, Univ Montpellier, CIRAD, INRAE, IRD, Montpellier SupAgro, Montpellier, France
[19]LMI IESOL, Centre IRD-ISRA de Bel Air, BP1386, CP18524 Dakar, Senegal
[20]Basque Centre for Climate Change, Sede Building 1, Scientific Campus of the University of the Basque Country, 48940, Leioa, Spain
[21]Ikerbasque, Basque Science Foundation, 48013 Bilbao, Spain
[22]University of Goettingen, Bioclimatology, Büsgenweg 2, 37077 Göttingen, Germany
[23]Institute of BioEconomy, National Research Council, 50145 Firenze, Italy
[24]Department of Ecology, University of Innsbruck, Innsbruck, Austria
[25]CSIRO Land & Water, Canberra, ACT, Australia
[26]School of Earth and Environmental Sciences, University of Queensland, St Lucia, 4067, Qld, Australia

*Contribute equally

Correspondence to: Yuan ZHANG (yuan.zhang@lmd.jussieu.fr)

40





**Abstract.** Aerosol and cloud-induced changes in diffuse light have important impacts on the global land carbon cycle by changing light distribution and photosynthesis in vegetation canopies. However, this effect remains poorly represented in current land surface models. Here we add a light partitioning module and a new canopy light transmission module to the ORCHIDEE land surface model (trunk version, v5453) and use the revised model, ORCHIDEE_DF, to estimate the fraction of diffuse light and its effect on gross primary production (GPP) in a multi-layer canopy. We evaluate the new parameterizations using flux observations from 159 eddy covariance sites over the globe. Our results show that compared to the original model, ORCHIDEE_DF improves the GPP simulation under sunny conditions and captures the observed higher photosynthesis under cloudier conditions in most plant functional types (PFTs). Our results also indicate that the larger GPP under cloudy conditions compared to sunny conditions is mainly driven by increased diffuse light in the morning and in the afternoon, and by decreased VPD and air temperature at midday. The observations show strongest positive effects of diffuse light on photosynthesis are found in the range 5-20 ℃ and VPD<1 kPa. This effect is found to decrease when VPD becomes too large, or temperature falls outside that range likely because of increasing stomatal resistance to leaf $CO_2$ uptake. ORCHIDEE_DF underestimates the diffuse light effect at low temperature in all PFTs and overestimates this effect at high temperature and high VPD in grasslands and croplands. The new model has the potential to better investigate the impact of large-scale aerosol changes on the terrestrial carbon budget, both in the historical period and in the context of future air quality policies and/or climate engineering.

# 1 Introduction

Process-based Land Surface Models (LSMs), which simulate the water and energy balance, and biogeochemical processes on land, have been widely used to attribute past changes in carbon (C) fluxes (Piao et al., 2009; Sitch et al., 2013) and to project the future land C budget (Ciais et al., 2013). Despite being useful and widely applied tools, large uncertainties are a limitation of LSMs (Sitch et al., 2008). One of the sources of the uncertainties is the omission or oversimplification of important processes that affect primary production. For instance, the impacts of light quality on photosynthesis is not currently represented in most LSMs, limiting the possibility to predict the variability of the carbon budget driven by changes in the atmospheric aerosol load which may be triggered by volcanic eruptions or variation in air pollution levels.

It has been found by in situ observations that under the same light level, the increase of diffuse light fraction can enhance light use efficiency and ultimately photosynthesis, or gross primary production (GPP) (Gu et al., 2003; Niyogi et al., 2004; Misson et al., 2005; Alton, 2007a; Knohl and Baldocchi, 2008; Mercado et al. 2009; Oliphant et al., 2011; Kanniah et al., 2013; Williams et al., 2014; Cheng et al., 2015; Wang et al., 2018). Several mechanisms explaining this GPP enhancement have been proposed and tested. First, the more isotropic nature of diffuse light means that it penetrates deeper into the canopy to become available for photosynthesis of the lower canopy leaves, which would otherwise be shaded and light limited (Roderick et al., 2001; Urban et al, 2012). Second, the multi-directionality of diffuse light produces a more homogeneous distribution of radiation between sunlit and shaded leaves, enhancing the photosynthesis of upper canopy shaded leaves and limiting the waste





of energy in light-saturated sunlit leaves (Li et al., 2014; Williams et al., 2014). Third, higher diffuse light fraction is often accompanied with less stressing temperature and vapor pressure deficit (VPD) for photosynthesis. The covariance of these

environmental factors may also cause the GPP to increase under cloudier conditions, although not being a direct effect of diffuse light (Gu et al., 2002; Cheng et al., 2015; Li et al., 2014). Finally, plant LAI (leaf area index, the area of leaves per unit land area) maximum may get acclimated to the cloudier seasons, which also contributes to higher GPP (Williams et al., 2016). Currently, most process-based LSMs simulate leaf photosynthesis using equations and parameterizations derived from Farquhar et al. (1980) with different formulations of stomatal conductance, usually with stomatal closure under high VPD or

low relative humidity (Ball et al., 1987; Yin et al., 2009; Medlyn et al., 2011). These parameterizations calculate photosynthesis per unit LAI considering the stress from temperature, VPD and soil water, and then integrate it over the entire canopy volume. Therefore, the effects of temperature and VPD change under cloudier conditions have been usually implicitly considered in current LSMs (e.g. Zhang et al., 2019). However, for the sake of simplicity and computational efficiency and for the lack of diffuse light fraction data, most global LSMs assumed a single extinction coefficient for both direct and diffuse light (Sellers

et al., 1997; Sitch et al., 2008). These LSMs are therefore incapable to investigate the effect of diffuse light fraction changes on photosynthesis. This limit of LSMs is thought to cause considerable underestimation of land C sink after the eruption of Mount Pinatubo (le Quere et al., 2018).

To better simulate the diffuse light impacts, several earlier works have developed photosynthesis models that considers different light transmission of diffuse and direct radiation (Spitters, 1986; Leuning et al., 1995; de Pury and Farquhar, 1997).

Based on these models, a few studies have tried to address the influence of light quality on GPP in LSMs. Dai et al. (2004) introduced a two-big-leaf canopy model to simulate the effects of diffuse and direct radiation in the Common Land Model (CLM 2L). However, this model assumes a single-layer canopy and can therefore not simulate the vertical profile of leaf traits. A multilayer canopy model is more suitable to represent the vertical heterogeneity of leaf traits and radiation transfer (Alton et al., 2007b; Bonan et al., 2012). Differentiating sunlit and shaded leaves in a multilayer canopy LSM was firstly considered

in the Joint UK Land Environment Simulator (JULES) LSM (Alton et al., 2007a; Mercado et al., 2009). Using this version of JULES, Mercado et al. (2009) investigated the diffuse light effect and suggested that diffuse light fraction change enhanced the global land C sink during the 1960-1999 period by about a quarter. However, Mercado et al. (2009)'s model was only tested at two forest sites which cannot represent well global terrestrial ecosystems. Thus, there remains need to obtain well-evaluated LSMs that distinguish diffuse and direct light to test the results of Mercado et al. (2009), and to further investigate

the diffuse radiation effect of aerosols.

Here we introduce a modified version of the LSM ORCHIDEE (Organizing Carbon and Hydrology In Dynamic Ecosystems, Krinner et al., 2005), referred to as ORCHIDEE_DF, which uses a semi-empirical method to calculate the fraction of diffuse light (Weiss and Norman, 1985), and a process-based multilayer canopy light transmission model to simulate the effects of diffuse light fraction on photosynthesis (Spitters, 1986). We evaluated the GPP simulated by ORCHIDEE_DF and the same

version of the ORCHIDEE code without diffuse light (trunk version, v5453) using observations collected from 159 eddy covariance flux sites over 11 plant functional types (PFT) (Baldocchi et al., 2001). Using both model simulations and





observations at the flux sites, we also investigated the interactions between diffuse light fraction and biotic and abiotic factors on GPP, with the objective of understanding when and how much does light quality affect photosynthesis. Because diffuse light is expected to enhance photosynthesis of shaded leaves in deep canopy, we would also test whether the enhancement of
GPP due to diffuse radiation is larger in canopies with larger LAI and whether environmental factors such as temperature or VPD affect this enhancement from diffuse light.

## 2 Data and Methods

### 2.1 Model description

#### 2.1.1 Canopy light transmission and photosynthesis in the ORCHIDEE trunk

The ORCHIDEE_DF model is based on ORCHIDEE trunk version 5453 (updated in September 2018). A general description of the physical processes related to energy and water balance, vegetation dynamics and biogeochemical processes in ORCHIDEE can be found in Krinner et al. (2005). The ORCHIDEE trunk version 5453 brings a number of improvements, and photosynthesis parameters were recently re-calibrated against FLUXNET data (Baldocchi et al., 2001) and atmospheric $CO_2$ observations for the IPSL Earth System Model (IPSL-CM6) and the CMIP6 simulations.

The leaf-scale photosynthesis calculation in the ORCHIDEE trunk is based on the scheme of Yin and Struik (2009). This scheme is an adaptation of the biophysical model of Farquhar et al. (1980) with a specific parameterization of stomatal conductance. The Farquhar et al. model calculates assimilation ($A$) as the minimum of the Rubisco-limited rate of $CO_2$ assimilation ($Ac$) and the electron transport-limited rate of $CO_2$ assimilation ($Aj$):

$$A = \min\{Ac, Aj\} \tag{1}$$

Here $Ac$ is mainly affected by the maximum carboxylation capacity of Rubisco ($Vcmax$), which is temperature dependent (Yin and Struik., 2009), and the $CO_2$ concentration at the carboxylation site ($Cc$):

$$Ac = \frac{(Cc - \Gamma^*)Vcmax}{Cc + KmC(1 + O/KmO)} - Rd \tag{2}$$

where $\Gamma^*$ is the $CO_2$ compensation point in the absence of dark respiration ($Rd$). $KmC$ and $KmO$ are the Michaelis-Menten constants for $CO_2$ and $O_2$, $O$ is the $O_2$ concentration at the carboxylation site.

$Aj$ is calculated as a function of $Cc$ and electron transport rate (J):

$$Aj = \frac{J(Cc - \Gamma^*)}{4.5Cc + 10.5\Gamma^*} - Rd \tag{3}$$

Here $J$ is determined by a temperature-dependent maximum electron transport rate ($Jmax$) and the photosynthetic photons absorbed by leaves, calculated following Yin and Struik (2009). Due to the attenuation of photosynthetically active radiation (PAR) with depth in the canopy, $J$ also varies vertically. In addition, to account for the distribution of light and maximize the
assimilation, plants tend to allocate nitrogen unevenly in the canopy profile (Niinemets et al., 1997; Meir et al., 2002), resulting in a vertical gradient in enzyme concentration and consequently in $Vcmax$ and $Jmax$. The vertical heterogeneity of canopy photosynthetic properties highlights the need to represent the canopy in a multilayer way.





In order to simulate the vertical transmission and absorption of light within the canopy, ORCHIDEE trunk uses a multilayer canopy with a big leaf approximation in each layer. The canopy is geometrically divided into up to a maximum number of 20

layers depending on the leaf area index (LAI). The discretization is represented in Fig. 1a and the LAI at the interface of the layers are given by:

$$LAI\_c_i = 12 \times \frac{e^{0.15 \times (i-1)} - 1}{e^{0.15 \times 20} - 1} \tag{4}$$

where $LAI\_c_i$ is the cumulative LAI above layer $i$, ($1 \leq i \leq 20$) and the layers are numbered from top to bottom. It should be noted that 20 layers are only for canopies with total LAI larger than 12. The number of layers decreases with total LAI. For

instance, if the LAI is 2, only the first 10 layers are used to calculate the light distribution and photosynthesis (Fig. 1a).

Light transmission in the multilayer canopy is calculated using the Beer-Lambert law (Monsi and Saeki, 1953) without distinguishing direct and diffuse light. The downward shortwave radiation arriving at the top of canopy (TOC) layer $i$ ($I_i$) is:

$$I_i = I_0 e^{-k \times LAI\_c_i} \tag{5}$$

where $k$ is the light extinction coefficient, taken equal to 0.5. $I_0$ is the TOC downward shortwave radiation (W m$^{-2}$).

Because the radiation attenuation between one layer and the one just below is assumed to be due to leaf absorption, the absorbed radiation per leaf area at the top of layer $i$ ($Iabs_i$) can be estimated as in Saeki (1960):

$$Iabs_i = \frac{-dI}{dLAI\_c}|LAI\_c_i = kI_0 e^{-kLAI\_c_i} \tag{6}$$

Here we assume that all canopy layers are thin enough to neglect the difference in light absorption within each canopy layer. i.e. the absorbed radiation does not attenuate within each canopy layer and $Iabs_i$ is used for all leaves in layer $i$.

It should be noted that the radiation considered to calculate the $J$ term in Eq. (3) is not shortwave radiation in W.m$^{-2}$ but photosynthetic photon flux density (PPFD) in μmol m$^{-2}$s$^{-1}$. A translation from $Iabs_i$ to the absorbed PPFD per leaf area in canopy layer $i$ ($PPFDabs_i$) is thus needed. Currently, there is no standard definition of the wavelength range for shortwave radiation (e.g. Howell et al, 1982; Zhang et al., 2004; Chen et al., 2012). In ORCHIDEE trunk, shortwave radiation in W m$^{-2}$ is multiplied by a factor of 0.5 to calculate photosynthetically active radiation (PAR) in W m$^{-2}$, and then a quanta-to-energy

ratio of 4.6 mmol J$^{-1}$ is used to convert PAR into PPFD in μmol m$^{-2}$ s$^{-1}$.

ORCHIDEE accounts for a vertical gradient in enzyme concentration in canopy. Vcmax and Jmax are assumed in the model to be linearly related to photosynthetically active leaf nitrogen concentration (per leaf area) (Kattge et al. 2007). Meir et al. (2002) found a decreasing leaf nitrogen concentration, as well as *Vcmax* and *Jmax* with increasing canopy depth in different ecosystems, suggesting an acclimation of plants to maximize photosynthesis in a canopy with unevenly distributed radiation.

ORCHIDEE trunk lacks an explicit model of dynamic N allocation to leaves in the canopy, instead, it uses an empirical relationship to represent the impact of leaf nitrogen concentration on *Vcmax* and *Jmax* using the vertical profile of radiation:

$$Vcmax_i = Vcmax_0 (1 - 0.7 \times (1 - \frac{I_i}{I_0})) \tag{7}$$

$$Jmax_i = Jmax_0 (1 - 0.7 \times (1 - \frac{I_i}{I_0})) \tag{8}$$





It should be noted that in ORCHIDEE trunk, the leaf-scale assimilation variables (e.g. $Vcmax$) are also affected by the
instantaneous air temperature and the temperature of the last month which plants have adapted to. The calculation of $Cc$
depends on VPD and also on whether the vegetation follows the C3 or C4 photosynthesis pathway (Yin and Struik, 2009). For
simplicity, the near surface air temperature and humidity are used for the calculation of assimilation in all canopy layers.
Furthermore, there are 13 PFTs in ORCHIDEE (Table S1) and $Vcmax$ and $Jmax$ are PFT-dependent.

### 2.1.2 Light partitioning in ORCHIDEE_DF

The lack of light quality (diffuse light fraction) information in most forcing datasets is one of the main difficulties when
simulating the diffuse light effect. Here we partition the half-hourly downward PAR, which can be derived from the shortwave
radiation, into diffuse and direct components following the Weiss and Norman (1985) empirical equations. Compared with
another empirical method (Spitters et al., 1986), we found that this method reproduces better the observed diffuse light fraction
at the flux sites used in this study (Fig. 2, Fig. S1). The diffuse PAR fraction ($Fdf_{PAR}$) above the canopy is estimated as:

$$Fdf_{PAR} = 1 - \frac{PAR_{p,dr}}{PAR_p}(1 - \left(\frac{a-R}{b}\right)^{\frac{2}{3}}) \tag{9}$$

where $PAR_p$ and $PAR_{p,dr}$ are the potential total and direct PAR, i.e. the total and direct PAR which would arrive at land surface
under clear sky conditions. $a$ and $b$ are parameters, which take values of 0.9 and 0.7, and $R$ is the ratio of observed to potential
total downward shortwave radiation ($SW_{obs}$ and $SW_p$) reaching the top of the canopy:

$$R = \frac{SW_{obs}}{SW_p} \tag{10}$$

The potential downward shortwave radiation consists of potential downward PAR (visible, range 0.4-0.7μm) and potential
downward near-infrared radiation (NIR, range 0.7-5μm). Also the potential PAR and NIR are the sum of direct ($PAR_{p,dr}$, $NIR_{p,dr}$)
and diffuse ($PAR_{p,df}$, $NIR_{p,df}$) components, given by:

$$SW_p = PAR_p + NIR_p = PAR_{p,dr} + PAR_{p,df} + NIR_{p,dr} + NIR_{p,df} \tag{11}$$

A simple atmospheric light transfer model modified from Weiss and Norman (1985) is used to estimate potential radiation.
The potential direct PAR, $PAR_{p,dr}$ is calculated as:

$$PAR_{p,dr} = PAR_{TOA}\, e^{-0.185(p/p_0)m} \cos\theta \tag{12}$$

where $PAR_{TOA}$ is the PAR at top of atmosphere (TOA), $p$ and $p_0$ indicate the local and standard sea level air pressure, $m$ is the
optical air mass, calculated using the solar zenith angle $\theta$:

$$m = (\cos\theta)^{-1} \tag{13}$$

The potential diffuse TOC PAR, $PAR_{p,df}$ is assessed as:

$$PAR_{p,df} = 0.4(PAR_{TOA} \cos\theta - PAR_{p,dr}) \tag{14}$$

which expresses that 40% of the PAR flux that is extinguished in the atmosphere through scattering and absorption is available
as diffuse PAR at the surface. Similarly, the potential direct and diffuse NIR at the top of the canopy ($NIR_{p,dr}$ and $NIR_{p,df}$
respectively), can be estimated as:





$$NIR_{p,dr} = (NIR_{TOA}e^{-0.06(p/p_0)m} - \omega)\cos\theta \qquad (15)$$

$$NIR_{p,df} = 0.6(NIR_{TOA}\cos\theta - NIR_{p,dr} - \omega\cos\theta) \qquad (16)$$

where $\omega$ is a flux term accounting for atmospheric water vapor absorption, calculated as a function of the solar constant ($SC$, in Wm$^{-2}$) and $m$:

$$\omega = SC \times 10^{(-1.195+0.4459\log_{10}m-0.0345(\log_{10}m)^2)} \qquad (17)$$

Using the results from Eqs. (12, 14, 15 and 16), we are able to calculate the $SW_p$ to obtain the value of $R$ in Eq. (10).

It should be noted that the quanta-to-energy ratio (in mmol J$^{-1}$) is different under different sky conditions, because atmospheric scattering varies spectrally with the air mass and the cloud amount (Dye, 2004). For this consideration, the calculation of PPFD from PAR in ORCHIDEE_DF uses the observation-oriented empirical equations from Dye (2004):

$$\beta_t = 4.576 - 0.03314 Fdf_{PAR} \qquad (18)$$

$$\beta_{df} = \frac{4.5886 Fdf_{PAR}}{0.010773 + Fdf_{PAR}} \qquad (19)$$

where the $\beta_t$ is the quanta-to-energy ratio for the total PAR ($PAR_t$) at the top of the canopy, while $\beta_{df}$ is for its diffuse component ($PAR_{df}$):

$$PPFD_t = \beta_t PAR_t \qquad (20)$$

$$PPFD_{df} = \beta_{df} PAR_{df} \qquad (21)$$

The diffuse PPFD fraction ($Fdf_{PPFD}$) can thus be calculated as:

$$Fdf_{PPFD} = \frac{PPFD_{df}}{PPFD_t} = \frac{\beta_{df}}{\beta_t} Fdf_{PAR} \qquad (22)$$

**2.1.3 Canopy light transmission in ORCHIDEE_DF**

In ORCHIDEE_DF, we use the same stratification of canopy as in the trunk version (Eq. (4)). But for the light transmission, we use a two-stream radiative transfer model following Spitters (1986). For convenience, we use radiation and $I$ in this section to refer to the PPFD derived from the light partitioning step.

An assumption of the model is that leaves are bi-Lambertian surfaces for radiation, i.e. the reflection and transmission are isotropic. This reflection and transmission are together referred to as leaf scattering. This assumption implies that once direct radiation encounters a leaf, it gets either absorbed or scattered as diffuse light. While for diffuse radiation, the scattered light remains diffuse. The scattering coefficient, σ, is assumed equal to 0.2 following Spitters (1986), meaning 20% of the light encountering a leaf is scattered (80% is absorbed).

Based on this assumption, the radiation penetrating the canopy can be divided into three components (Fig. 3): the direct light which has not been intercepted by leaves ($I_{dr,dr}$), the diffuse light generated by leaf scattering of intercepted direct light ($I_{dr,df}$), and the diffuse light in the canopy provided by the TOC diffuse radiation ($I_{df}$). It should be noted that the diffuse light generated by multiple times of scattering of the direct light is grouped into $I_{dr,df}$, while those from the scattering of TOC





diffuse radiation belong to $I_{df}$ (Fig. 3). The sum of $I_{dr,dr}$ and $I_{dr,df}$ hereafter noted as $I_{dr}$ represents the total radiation in each

canopy layer derived from the TOC direct radiation, hereafter $I_{dr,0}$.

If we also consider direct radiation as parallel beams, only the first leaves on the way of direct light can absorb $I_{dr,dr}$. These

leaves are referred to as sunlit leaves. The fraction of sunlit leaves in each canopy layer can be calculated by applying Beer-

Lambert law using an extinction coefficient for opaque, non-reflective "black" leaves (Fig. 1b):

$LAIf_{sun,i} = e^{-k_b LAI\_c_i}$                                                                           (23)

here $LAIf_{sun,i}$ is the fraction of sunlit LAI in canopy layer i. $LAI\_c_i$ is the cumulative LAI in Eq. (4). $k_b$ is the extinction

coefficient if the leaves are assumed "black". A function of $\theta$, leaf angle distribution index (*LA*) and leaf clumping index (*LC*)

is used to represent the geometry between the direct radiation and leaves:

$k_b = \frac{LA*LC}{\cos\theta}$                                                                                     (24)

For spherically distributed leaves, *LA* equals 0.5 (Goudriaan, 1977; Bodin and Franklin, 2012). *LC* is defined as in Myneni et

al. (1989) and Baldocchi and Wilson (2001), varying between 0 and 1. Here we use the value 0.85 instead of 0.84 as

recommended by an observationally-based study (Baldocchi and Wilson, 2001).

The leaves which cannot absorb $I_{dr,dr}$ are referred to as shaded leaves. The fraction of shaded LAI in canopy layer i ($LAIf_{shd,i}$)

is thus the complement of $LAIf_{sun,i}$:

$LAIf_{shd,i} = 1 - LAIf_{sun,i}$                                                                          (25)

Because $I_{dr,dr}$ is assumed not to be transmitted as direct radiation through leaves, $I_{dr,dr,i}$, which represents $I_{dr,dr}$ at layer i can

be calculated similarly as in Eq. (23) using the downward direct radiation at the top of the canopy ($I_{dr,0}$):

$I_{dr,dr,i} = I_{dr,0} e^{-k_b LAI\_a_i}$                                                                      (26)

The transmission of $I_{dr,df}$ is difficult to estimate directly. Here we calculate it as the difference between $I_{dr}$ and $I_{dr,dr}$ in each

layer:

$I_{dr,df,i} = I_{dr,i} - I_{dr,dr,i}$                                                                               (27)

where $I_{dr,df,i}$ and $I_{dr,i}$ represent net (downward minus upward) $I_{dr,df}$ and net $I_{dr}$ at layer i, respectively.

The calculation of $I_{dr,i}$ is based on Goudriaan (1982) and Hikosaka et al. (2016) under the assumptions that there is no

difference in optical traits between leaves from different canopy layers and that the canopy is deep enough to neglect the

reflection of the soil:

$I_{dr,i} = (1-\rho)I_{dr,0} e^{-\sqrt{1-\sigma}k_b LAI\_a_i}$                                                             (28)

where $\rho$ indicates the canopy reflection coefficient (i.e., the ratio between the TOC downward and upward radiation),

calculated as:

$\rho = (\frac{1-\sqrt{1-\sigma}}{1+\sqrt{1-\sigma}})(\frac{2}{1+1.6cos\theta})$                                                           (29)





In contrast to the direct light transmission, the diffuse light will not change its directional characteristics when scattered by leaves. Similar to Eq. (5), net $I_{df}$ at canopy layer $i$ ($I_{df,i}$) can be estimated using TOC downward diffuse radiation ($I_{df,0}$) in a Beer-Lambert equation:

$$I_{df,i} = (1 - \rho)I_{df,0}e^{-k_d LAI\_a_i} \tag{30}$$

where $k_d$ is the light extinction coefficient for diffuse light, calculated following Spitters (1986) as:

$$k_d = 0.8\sqrt{1-\sigma} \tag{31}$$

Similar to Eq. (6), the flux of light that is absorbed per canopy leaf area in layer $i$ from $I_{df}$ ($Iabs_{df,i}$), $I_{dr}$ ($Iabs_{dr,i}$), and $I_{dr,dr}$ ($Iabs_{dr,dr,i}$) can be written respectively as:

$$Iabs_{df,i} = \frac{-dI_{df}}{dLAI\_c}|LAI\_c_i = k_d I_{df,i} \tag{32}$$

$$Iabs_{dr,i} = \frac{-dI_{dr}}{dLAI\_c}|LAI\_c_i = \sqrt{1-\sigma}k_b I_{dr,i} \tag{33}$$

$$Iabs_{dr,dr,i} = \frac{-dI_{dr,dr}}{dLAI\_c}|LAI\_c_i = k_b I_{dr,dr,i} \tag{34}$$

The $I_{dr,df}$ absorbed per canopy leaf area by layer i ($Iabs_{dr,df,i}$) is:

$$Iabs_{dr,df,i} = Iabs_{dr,i} - Iabs_{dr,dr,i} \tag{35}$$

It should be noted that all leaves can absorb diffuse radiation. Therefore Eq. (32) and Eq. (35) also represent the absorption of $I_{df}$ and $I_{dr,df}$ at the leaf scale. However, $I_{dr,dr}$ is only absorbed by sunlit leaves, thus the absorption of $I_{dr,dr}$ per sunlit leaf

area does not equal to $Iabs_{dr,dr,i}$, which is the average at canopy scale. Instead, because $I_{dr,dr}$ does not change its intensity, the absorption of $I_{dr,dr}$ per sunlit leaf area can be written as:

$$Iabs_{dr,dr,i,sun} = (1-\sigma)k_b I_{dr,0} \tag{36}$$

We have assumed that shaded leaves can only absorb diffuse light. Then, the radiation absorbed (per leaf area) by shaded leaves layer i ($I_{shd,i,abs}$) is:

$$Iabs_{shd,i} = Iabs_{df,i} + Iabs_{dr,df,i} \tag{37}$$

The sunlit leaves also absorb the direct light besides diffuse light. The radiation received by sunlit leaves can thus be calculated as:

$$Iabs_{sun,i} = Iabs_{shd,i} + Iabs_{dr,dr,i,sun} \tag{38}$$

Apart from light transmission, all other parameters (e.g. *Vcmax*, *Jmax*) in ORCHIDEE_DF are kept the same as in ORCHIDEE

trunk.

### 2.2 Flux data and site level simulations

To evaluate ORCHIDEE_DF, we collected flux site measurements from the La Thuile dataset, which includes 965 site-year observations from in total 252 sites (https://fluxnet.fluxdata.org/data/la-thuile-dataset/). Because our ORCHIDEE simulations assume that the ecosystems are in equilibrium and do not experience disturbances (e.g., logging, fire), we selected flux sites





without strong disturbances during the last 10 years. For sites that also provided growing season LAI information, we also removed forests site with LAI<2, which may be considered as sparse forests with understory vegetation. In the end, observations of 655 site-years from 159 sites were retained (Table S2). The annual climate of the sites spans from -9 ºC to 27ºC in temperature, and from 67 mm yr$^{-1}$ to over 3000 mm yr$^{-1}$ in precipitation (Fig. S2), which is representative to most of the climate conditions over the globe. The dataset provides in situ meteorology, net ecosystem exchange (NEE), gross primary

productivity (GPP), and data quality information at 30-min time steps. The GPP provided by this dataset is partitioned from NEE and gap filled using the method of Reichstein et al. (2005). Specifically, 64 of the 159 sites provided measurements of both total and diffuse PPFD, which allows us to evaluate the light partitioning parametrization (Eqs. (9-20)). The gaps and missing variables in meteorology are filled using the approach from Vuichard and Papale (2015) to meet the model input requirements.

Because ORCHIDEE has different photosynthesis parameters for different PFTs, we classified the vegetation at each site into the 13 ORCHIDEE PFTs (Table S1) according to the IGBP land cover types specified on the website of FluxNet ([www.fluxdata.org](www.fluxdata.org)). If the IGBP land cover type is not specified or may match more than one ORCHIDEE PFTs (e.g. shrublands, savannas and wetlands), the PFT is determined according to the dominant plant species described in related references. Specifically, the mixed forests (MF) type exists in the IGBP classification but not in the ORCHIDEE PFTs. Because

MF sites are mostly located in temperate regions, we assume that they are composed of 50% temperate broadleaf deciduous forests and 50% temperature evergreen needle-leaf forests. Detailed information of flux sites is found in Table S2.

To evaluate the model, spinup simulations of 30 years are firstly conducted on ORCHIDEE_DF at each site to equilibrate the leaf area index with site conditions. Then the simulations with 30 min output are conducted with ORCHIDEE trunk and ORCHIDE_DF, using the full span of the Fluxnet la Thiule series respectively at each site. It should be noted that we use the

same spinup for ORCHIDEE trunk and ORCHIDEE_DF to ensure the same initial states of the two simulations. A test has shown that different spinup simulations do not affect the simulation of GPP in the following years (not shown).

## 2.3 Analyses

When evaluating the modeled GPP response to diffuse light we have not used all the 30-min data points due to several concerns. First, all night time data points were excluded from the analyses given that GPP is zero at night. Second, all data points flagged

with poor quality in the FLUXNET archive have been removed. Third, ORCHIDEE might not be perfect in capturing the seasonality of leaf flushing and shedding. In order to minimize the uncertainty from phenology, we used only data from the growing season at each site, which is defined as months when:

$$GPP_m > GPP_{m,min} + (GPP_{m,max} - GPP_{m,min})/4 \qquad (39)$$

where $GPP_m$ is the observed monthly GPP, $GPP_{m,min}$ and $GPP_{m,max}$ are the observed minimum and maximum monthly GPP at

the corresponding sites.

To assess the effect of variable diffuse light fraction on both GPP and light use efficiency (LUE, the ratio of GPP to incoming shortwave radiation), we look at the difference in GPP and LUE during sunny and cloudy conditions. We define sunny and





cloudy conditions as those when the fraction of diffuse PPFD at the top of the canopy ($Fdf_{PPFD}$) is smaller than 0.4 and greater than 0.8, respectively, and calculate the average sunny and cloudy GPP and LUE at each site. To ensure that the comparison

between sunny and cloudy conditions are at the same PPFD level, the sunny time steps with PPFD larger than the maximum PPFD under cloudy conditions are removed from the average, and vice versa. In addition, to make sure that the difference in GPP between sunny and cloudy is not an artifact of different LAI, sites with average modelled LAI under cloudy and sunny conditions differing by more than 0.3 are excluded from this analysis.

## 3 Results

### 3.1 Diffuse light modeling

Fig. 2 shows the relationship between 30-min modeled and measured $Fdf_{PPFD}$ at flux sites (64 sites). The data points are generally distributed along the 1:1 line, indicating an unbiased estimation of our diffuse light model. In total, our simple model explains over 51% of the variance in observed diffuse PPFD fraction. Although this model is imperfect, we currently have no better way to reproduce the diffuse PPFD at the flux site scale.

### 3.2 General model performance

The performance of both ORCHIDEE trunk and ORCHIDEE_DF for 30-min GPP from each PFT (all sites) is presented in Fig. 4. Generally, ORCHIDEE trunk underestimated the standard deviation (STD) of GPP at 30-min time-step compared with observations, and across all PFTs except Boreal evergreen needleleaf forests and C4 Croplands (Fig. 4a). The correlation coefficients between ORCHIDEE trunk GPP and observations are generally between 0.5 and 0.7 among PFTs (Fig. 4b). In

tropical broadleaf forests, this correlation coefficient is about 0.2, which is much smaller than in other PFTs and likely due to the limited seasonality of primary production in the tropics. The GPP simulated by ORCHIDEE_DF shows comparable performance with ORCHIDEE trunk, but with slightly smaller STD (Fig. 4a).

Similar evaluations on the GPP from the two models are performed under cloudy and sunny conditions respectively (Fig. 4c-f). Under cloudy conditions, ORCHIDEE trunk and ORCHIDEE_DF both underestimated GPP STD. The correlation

coefficients to observations are generally between 0.5 and 0.8 (Fig. 4d). Compared with ORCHIDEE trunk, ORCHIDEE_DF shows slightly worse correlation coefficients but improves STD for most of the PFTs except Tropical broad-leaved evergreen forests (TrEBF) and Temperate needleleaf evergreen forests (TeENF) (Fig. 4c).

Compared with cloudy conditions, the GPP simulated by the two models under sunny conditions show weaker correlation to observations. The correlation coefficients generally vary between 0.3 and 0.6 among PFTs. However, it should be noted that

ORCHIDEE_DF better reproduced GPP variation under sunny conditions compared with ORCHIDEE trunk in most PFTs except TeDBF and C4Cro (Fig. 4f). The GPP STD derived from ORCHIDEE trunk simulations under sunny conditions show larger variability among PFTs than under cloudy conditions. While for ORCHIDEE_DF, the GPP STD under sunny and cloudy conditions show similar bias compared with observations (Fig. 4e).





### 3.3 Effects of diffuse light on GPP and LUE

Because the modification of ORCHIDEE_DF was limited to light transmission, the pertinent process-oriented evaluation of the two models should focus on their ability to capture the observed GPP differences between cloudy and sunny conditions (hereafter ΔGPP), rather than on correlations or RMSE with observations, that may result from different structural and parametric errors of the model, not related to diffuse light.

    Figure 5 shows the observed and modeled GPP under sunny and cloudy conditions at different PPFD levels at flux sites with

relatively long time series of observations from each PFT. For all the sites selected, the observed GPP under cloudy conditions is larger than under sunny conditions. However, the GPP simulated by ORCHIDEE trunk shows no or small difference between cloudy and sunny conditions at most sites. In contrast, ORCHIDEE_DF reproduces this GPP difference in most PFTs except TrDBF, BoDBF and C4Gra. However, there is only one TrDBF site and very few C4Gra sites in our dataset. Furthermore, at most C4Gra sites, we are not able to find PPFD levels where sunny and cloudy conditions co-exist. Therefore, we are not able

to make further evaluation of cloudy-minus-sunny GPP differences for TrDBF and C4Gra. At three of the four BoDBF sites, the modeled GPP difference under cloudy and sunny conditions is relatively small (not shown). This might be because the model overestimated the deleterious effect of low temperature on photosynthesis at the BoDBF sites (mean annual temperature<3ºC). In total, observations from about 70% of the sites show remarkable higher GPP under cloudy than sunny conditions. This percentage is only 30% in ORCHIDEE trunk simulations but 60% in ORCHIDEE_DF simulations.

To summarize the site level results, we investigated the distribution of GPP difference between cloudy and sunny conditions (here after refer to as ΔGPP) (Fig. 6a). Observations and ORCHIDEE_DF show a positive bias in ΔGPP, with ΔGPP values between 0-3×10$^{-4}$ gC m$^{-2}$ s$^{-1}$ at most sites. However, for ORCHIDEE trunk, ΔGPP is near zero at most sites. This result confirms that ORCHIDEE_DF performs much better than ORCHIDEE trunk in simulating differences in GPP under different light conditions.

It should be noted that ΔGPP can be affected by PPFD. At sites where sunny and cloudy conditions only coexist at a relatively low PPFD level, the ΔGPP should be also small. To remove the effect of PPFD level on ΔGPP, we analyzed the difference in LUE, i.e. ΔLUE, between the two conditions (Fig. 6b). Compared with ΔGPP, positive ΔLUE values are more evenly distributed around 0-15×10$^{-8}$ gC μmol$^{-1}$ photon for observation and ORCHIDEE_DF simulation. For ORCHIDEE trunk, the ΔLUE has the range of 0-8 ×10$^{-8}$ gC μmol$^{-1}$ photon, with the upper range smaller than in the observations and ORCHIDEE_DF.

We further refined this analysis to investigate if the effects of diffuse light differ at different times of the day (Fig. 7). Results for three different periods in the day show that in the morning and afternoon, cloudy conditions result in higher GPP of 0-5×10$^{-4}$ gC m$^{-2}$ s$^{-1}$ than sunny conditions at most sites, which is generally captured by ORCHIDEE_DF but missed by ORCHIDEE trunk in the morning (Fig. 6a, c). At midday, due to the dependence of Fdf on PPFD (Eqs. (9 and 10)), we fail at many sites to find PPFD levels where sunny and cloudy conditions coexist. Nevertheless, the result generally indicates larger

mid-day ΔGPP than those in the morning and afternoon. It should be noted that this large difference is captured by both ORCHIDEE_DF and ORCHIDEE trunk (Fig. 7b). Because direct and diffuse light are not distinguished in ORCHIDEE trunk,





this midday ΔGPP should be mainly contributed by environmental factors other than diffuse light fraction. The ΔLUE derived by ORCHIDEE_DF also shows a largely similar distribution as in observations, but ORCHIDEE trunk underestimates the morning and afternoon ΔLUE (Fig. 7d-f).

### 3.4 Interactions between diffuse light and environmental factors

As implied by Fig. 7, the diffuse light fraction is not the only factor causing ΔGPP. Other possible factors include temperature and VPD (Gu et al., 2002; Cheng et al., 2015; Li et al., 2014). Here, we thus investigate the diffuse light effect along temperature and VPD gradients in Fig. 8. To remove the effect of PPFD level, we only show ΔLUE.

ΔLUE shows a unimodal curve along the temperature gradient for observation and the two models (Fig. 8a). At low
temperature, both models indicate a very low ΔLUE of 1 gC μmol$^{-1}$ photon, which is about 1/3 of the ΔLUE derived from observations. With increasing temperature, the observed ΔLUE shows a maximum at 10-20 °C, with a magnitude of ~$8 \times 10^{-8}$ gC μmol$^{-1}$ photon and declines slightly at higher temperatures. The peak of ΔLUE simulated by ORCHIDEE_DF has a magnitude comparable to that of observations, but at higher temperature (20-25 °C) than for observations. The ΔLUE simulated by ORCHIDEE trunk is much smaller, with a peak of ~$4 \times 10^{-8}$ gC μmol$^{-1}$ photon at 10-15 °C.

The effect of VPD on ΔLUE is shown in Fig. 8b. For observations and both model simulations, a monotonic decreasing trend of ΔLUE along the VPD gradient is found. The ΔLUE from observations and ORCHIDEE_DF show a comparable magnitude, from $8 \times 10^{-8}$ gC μmol$^{-1}$ photon at VPD<0.5 kPa to $5 \times 10^{-8}$ gC μmol$^{-1}$ photon at 2-4 kPa VPD level. The ΔLUE simulated by ORCHIDEE trunk is smaller than observations.

Apart from environmental factors, the effects of diffuse light may also differ among PFTs because different PFTs have different
canopy structures and photosynthetic parameters (e.g. Vcmax). Here we analyzed the ΔLUE in forests and short vegetation (grasslands and croplands) separately (Fig. 8c-f). In forests (Fig. 8c, d), ORCHIDEE_DF underestimates ΔLUE at temperatures lower than 20 °C. It also largely captures the observed ΔLUE trend with VPD, while ORCHIDEE trunk underestimates ΔLUE at all cases. Compared with forests, in short vegetation (Fig. 8e, f), observations show a stronger decline of ΔLUE at high temperatures (>25 °C) and high VPD conditions (>0.5 kPa). However, for ORCHIDEE_DF, the short vegetation ΔLUE
remains as high as for forests.

Fig. 9 shows the distribution of ΔLUE in the Temperature-VPD dimensions. Observations indicate that the largest ΔLUE is reached under conditions when temperature is in the range 5-20 °C and VPD <1 kPa (Fig. 9a). This temperature is thought more favorable for photosynthesis as it is generally consistent with the photosynthesis optimum temperature detected by Huang et al. (2019) in latitudes where most of the sites are located. Under these conditions, the ΔLUE is usually over $7 \times 10^{-8}$ gC μmol$^{-}$
$^1$ photon. When the temperature is lower than 5 °C or higher than 20 °C, or VPD becomes larger than 1 kPa, ΔLUE tends to decline. Compared with observations, the ΔLUE simulated by ORCHIDEE_DF shows a similar decreasing trend with VPD at all temperature levels (Fig. 9c), however, no obvious decline of ΔLUE is found at high temperatures. The ΔLUE simulated by ORCHIDEE trunk is much smaller compared with observations (Fig. 9b).





The ΔLUE from forests and short vegetation are shown separately in Fig. 10. Based on site level observations (Fig. 10a), both
vegetation types show a larger ΔLUE at lower VPD between 5-20 °C. In forests, there is also large ΔLUE at high temperature
conditions, which mainly occurs in tropical forests (Fig. S3). Nevertheless, ORCHIDEE_DF still overestimates the ΔLUE at
high temperatures (Fig. 10e), which is mainly due to the overestimation of ΔLUE at high temperatures for temperate forests
(Fig. S3).

Compared with forests, the short vegetation shows a much stronger decline of ΔLUE at higher VPD level (Fig. 10b), however,
it is not well captured by ORCHIDEE_DF (Fig. 10f). In most cases, ORCHIDEE trunk tends to strongly underestimate ΔLUE
unless the observed ΔLUE is small or negative (e.g. VPD > 2kPa for short vegetation).

## 4 Discussion

### 4.1 Improvement of ORCHIDEE_DF

The role of diffuse light on photosynthesis has been found and modeled in different vegetation types (Gu et al., 2003; Niyogi
et al., 2004; Misson et al., 2005; Alton et al., 2007b; Knohl and Baldocchi, 2008; Mercado et al. 2009; Oliphant et al., 2011;
Kanniah et al., 2013; Williams et al., 2014; Cheng et al., 2015; Wang et al., 2018). However, very few studies have attempted
to account for the diffuse light effect in a global land surface model, and fewer studies have used large FLUXNET datasets for
evaluation. Here, by using flux observations from 159 sites over the globe, we show that by separating the direct and diffuse
light, ORCHIDEE_DF improves the simulation of GPP under sunny conditions and, more importantly, reproduced the
observed impacts of diffuse light on GPP and LUE for most of the PFTs (Figs. 4-6). Under cloudy conditions, ORCHIDEE_DF
seems to perform slightly worse than ORCHIDEE trunk (Fig. 4). However, it should be noted that ORCHIDEE_DF has not
been recalibrated and all parameters are those from ORCHIDEE trunk despite the substantial changes in the code with respect
to light partitioning and canopy light transmission. Furthermore, the GPP simulated by ORCHIDEE trunk shows different GPP
STD biases under sunny and cloudy conditions, while ORCHIDEE_DF gives a more systematically underestimated GPP STD,
which should be more easily corrected in a future calibration. The site level comparison (Fig. 5) also explains how
ORCHIDEE_DF reproduces the GPP increase compared to ORCHIDEE trunk. At most sites, the GPP simulated by the two
models show similar magnitude under cloudy conditions. While under sunny conditions, the GPP simulated by
ORCHIDEE_DF is significantly smaller. This is because in the one-stream canopy light transmission model in ORCHIDEE
trunk, all light is considered as diffuse light and evenly distributed in each leaf layer. This simplified approach to the modelling
of light distribution leads to larger GPP under sunny conditions because the effect of light saturation on sunlit leaves is ignored.
Since ORCHIDEE trunk was calibrated using both sunny and cloudy data, but ORCHIDEE_DF corrected the overestimation
under sunny conditions, ORCHIDEE_DF may give an overall underestimation using current parameters.





## 4.2 Factors affecting the response of GPP to diffuse light

Although diffuse light can increase photosynthesis of shaded leaves, the GPP increase under cloudy conditions is not
contributed only by this effect. A recent field study suggested that photosynthesis from part of the canopy (especially sunlit
leaves) benefits from the lower VPD rather than the higher diffuse light fraction under cloudier conditions (Wang et al., 2018).
Our results show that during the morning and the afternoon, higher diffuse PAR fraction is the main factor causing larger GPP
under cloudy conditions compared with sunny conditions, as only ORCHIDEE_DF reproduced the observed positive ΔGPP
during the two periods (Fig. 7). While at midday, the larger GPP under cloudy conditions should be mainly due to lower T or
VPD other than to diffuse light because ORCHIDEE trunk, which does not simulate the diffuse light effect, also reproduces
this effect (Fig. 7). A similar effect is also reported by Cheng et al. (2015), who found that in croplands the midday GPP
increase under cloudier conditions is mainly caused by lower temperature and lower VPD rather than by diffuse light.
Photosynthesis is often considered as limited by either carboxylation or electron transportation (Farquhar et al., 1980). It is
when the shaded leaf photosynthesis is limited by light that diffuse light can increase GPP. At midday, large VPD may cause
stomatal closure, leading to a carboxylation-limited photosynthesis. Our results imply that it might be important to consider
the diurnal cycle of environmental factors to better understand the effect of diffuse light.

It should be noted that the covariation of environmental factors with more diffuse light under cloudier conditions does not
always benefit photosynthesis. For instance, if the vegetation is cold stressed under cooler conditions, the decrease of
temperature under cloudier condition may strengthen this stress and offset the effect of diffuse light. Our analyses indicate that
under most stressed conditions, the effect of diffuse light on photosynthesis is weakened (Figs. 9, 10).

Another important factor is the light itself. When there is no light saturation of shaded leaves, under the same diffuse light
fraction, stronger light levels are likely to benefit the shaded leaves more, resulting in higher ΔGPP (Fig. 5, ΔGPP tends to be
larger at higher PPFD level at most sites). Nevertheless, apart from GPP, in this study we also investigated LUE (the
photosynthesis per unit PPFD), which has removed this effect.

Besides environmental factors, canopy structure is also very important. Theoretically, thicker canopies with large LAI tend to
be more sensitive to diffuse light because a larger fraction of leaves are light limited due to shading (Fig. 1). As expected,
ORCHIDEE_DF has shown an increasing ΔLUE with LAI (Fig. 11). However, the analyses based on LAI observations
suggested a very weak positive effect of LAI on ΔLUE (Fig. 11). This insensitive response of ΔLUE to LAI detected here
should be treated with caution because the LAI observations are not well defined (maximum or average) and remain very
limited in the current FLUXNET dataset (less than 10 in each LAI interval). Using more detailed LAI and $CO_2$ flux
observations, Wohlfahrt et al. (2008) has clearly exhibited the influence of LAI on diffuse light-induced photosynthesis
changes at a grassland.



### 4.3 Uncertainty and Limitations

Many empirical methods have been proposed to partition solar radiation into diffuse and direct light (e.g. Spitters et al., 1986;
Weiss and Norman, 1985; Erbs et al., 1982). However, biases remain in the predicted diffuse light fraction under all aerosol
and cloud conditions, which inevitably introduce some uncertainties to our analyses. Nevertheless, such methods are currently
the most feasible approach at flux site level. More continuous measurements of direct and diffuse surface radiation at more
sites are desirable.

Another source of uncertainty is from the light transmission model. In ORCHIDEE_DF, we used a two-stream radiative
transfer approximation. In this model, the canopy trait parameters such as leaf scattering, leaf orientation and leaf clumping
factors are assumed the same for all PFTs, however real canopies are very diverse (Smith et al., 2004). In situ observations are
required to obtain better parameters. Furthermore, the validity of the light transmission model in ORCHIDEE_DF depends on
the several assumptions described in the model description section. These assumptions are not always valid. For example,
because direct solar beams are not exact parallels, leaves in canopies are not always sunlit or shaded, they may also fall in
penumbra regions, (i.e. regions where only part of the incoming direct solar beams are blocked, Smith et al., 1989; Cescatti
and Niinemets, 2005). These more complex processes should be considered in future model development. Nevertheless, our
simplified light transmission already succeeds in reproducing the observed diffuse light impact.

There are other sources of uncertainties in complex land surface models. Although ORCHIDEE_DF reproduces the magnitude
of the diffuse light effects, it fails to reproduce the response of ΔLUE to temperature. For all PFTs, ORCHIDEE_DF
underestimates the ΔLUE at low temperatures, and overestimates ΔLUE at high temperatures (Fig. 8). The low temperature
underestimation is also found in ORCHIDEE trunk, indicating that the models may have underestimated the tolerance of plants
to low temperatures. While at high temperatures, ORCHIDEE_DF tends to underestimate the impact of heat stress. This bias
might be due to the parameterization of temperature acclimation which is based on observations mainly from a narrow
temperature range (11-29 ℃) (Kattge et al. 2007). For short vegetation, the introduction of diffuse light into the model results
in an increase of ΔLUE at high temperatures and high VPD (Figs. 8, 10), indicating the vegetation simulated by ORCHIDEE
trunk remains light limited under such conditions. However, the strong decreasing trend of observed ΔLUE along temperature
and VPD gradients indicates heat and VPD stress. This implies that parameters in current ORCHIDEE version may have
underestimated the response of grassland and cropland photosynthesis to heat and VPD stress.

### 5 Conclusion

In this study, we added to the ORCHIDEE trunk a module to partition the downward surface solar radiation into diffuse and
direct components, and a new canopy radiative transfer model, which separates the existing multilayer canopy into sunlit and
shaded leaves. The resulting new land surface model, ORCHIDEE_DF, is evaluated using the La Thuile flux dataset over 159
sites over 11 PFTs. Compared with ORCHIDEE trunk, ORCHIDEE_DF improves the GPP simulation under sunny conditions.
This improvement successfully reproduces the observed enhancement of GPP under cloudier conditions at most of the sites.





Using observed and modeled GPP, we found an increase of GPP under cloudier conditions at all times of the day; however, the mechanisms causing this effect are different at midday from morning and afternoon. During morning and afternoon, the increase in GPP is mainly caused by increased diffuse light fraction, while at midday, the GPP increase is mainly due to weaker stress from temperature and VPD.

Observations indicate that under cloudy and sunny conditions for the same light level, the maximum LUE difference can be

over $7 \times 10^{-8}$ gC μmol$^{-1}$ photon. The maximum LUE is found at temperature and VPD conditions more favorable for photosynthesis (5-20 ℃ for temperature and < 1 kPa for VPD). With increasing VPD, or under lower or higher temperatures, the LUE may decrease. Compared with observations, ORCHIDEE_DF underestimates the diffuse light effect at low temperature and overestimates it at high temperatures, possibly due to imperfect temperature acclimation parameterization in the current ORCHIDEE model. In grasslands and croplands, ORCHIDEE_DF overestimates the diffuse light effect on LUE,

which might be due to an overestimation of their tolerance to dry conditions.

As ORCHIDEE_DF is a land surface model which is able to capture the effect of diffuse light for a large number of sites over the globe, we are confident that, with this improved model framework and proper calibration, we can investigate the effect of aerosols on global biogeochemical cycles, and assess the impact of aerosol emission policies and aerosol related climate engineering on such cycles.

**Code and data availability**

The code of the ORCHIDEE_DF is available at https://forge.ipsl.jussieu.fr/orchidee/wiki/GroupActivities/CodeAvalaibilityPublication/ORCHIDEE_DFv1.0_site. Flux data (La Thuile) used in this study is available at https://fluxnet.fluxdata.org/data/la-thuile-dataset/.

**Acknowledgement**

The authors acknowledge support from European Research Council Synergy project SyG-2013-610028 IMBALANCE-P and the ANR CLAND Convergence Institute. The authors are very grateful to the FLUXNET communities for their efforts at making sites and collecting data, and specially to flux site PIs who are not in the author list but have given constructive suggestions on this manuscript. The authors also acknowledge Dr. Yves Balkanski and Dr. Nicolas Viovy for their suggestions during this work.

**Author contributions.**

PC, OB and LL designed the project. YZ developed the model code with help from AB, FM, DG and AC. NV provided the code for data gap filling. YZ prepared the paper with contributions from all the co-authors.





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





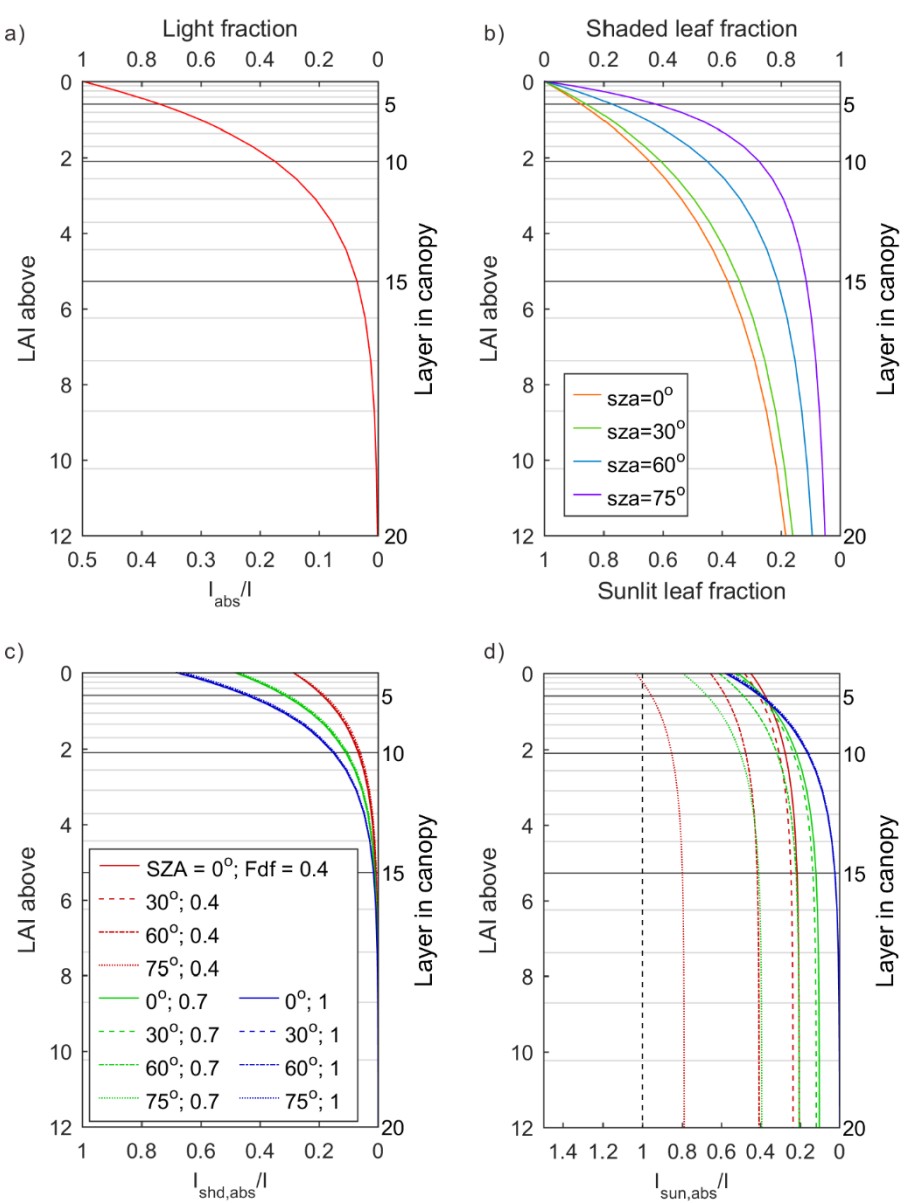

**Figure 1: The distribution of light and leaves in canopy. (a) light distribution in ORCHIDEE trunk. (b) distribution of sunlit and shaded leaves in canopy in ORCHIDEE_DF. (c) light absorbed by shaded leaves in each canopy layer under different solar zenith angle (SZA) and fraction of diffuse light (Fdf) in ORCHIDEE_DF. (d) Same as (c) but for sunlit leaves. I, downward PPFD at the top of the canopy; $I_{abs}$, PPFD absorption per leaf area in ORCHIDEE trunk; $I_{shd,abs}$, PPFD absorption per leaf area in shaded leaves; $I_{sun,abs}$, PPFD absorption per leaf area in sunlit leaves.**

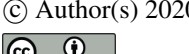



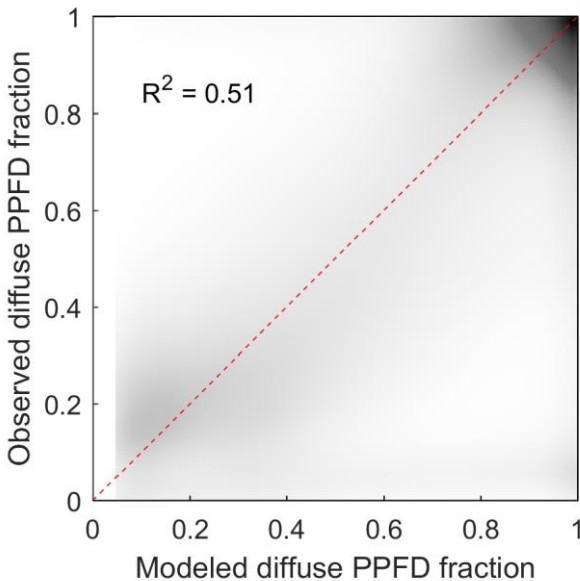

**Figure 2: Modeled and observed diffuse PPFD fraction. The dark area indicates high data density, while light area indicates low data density.**



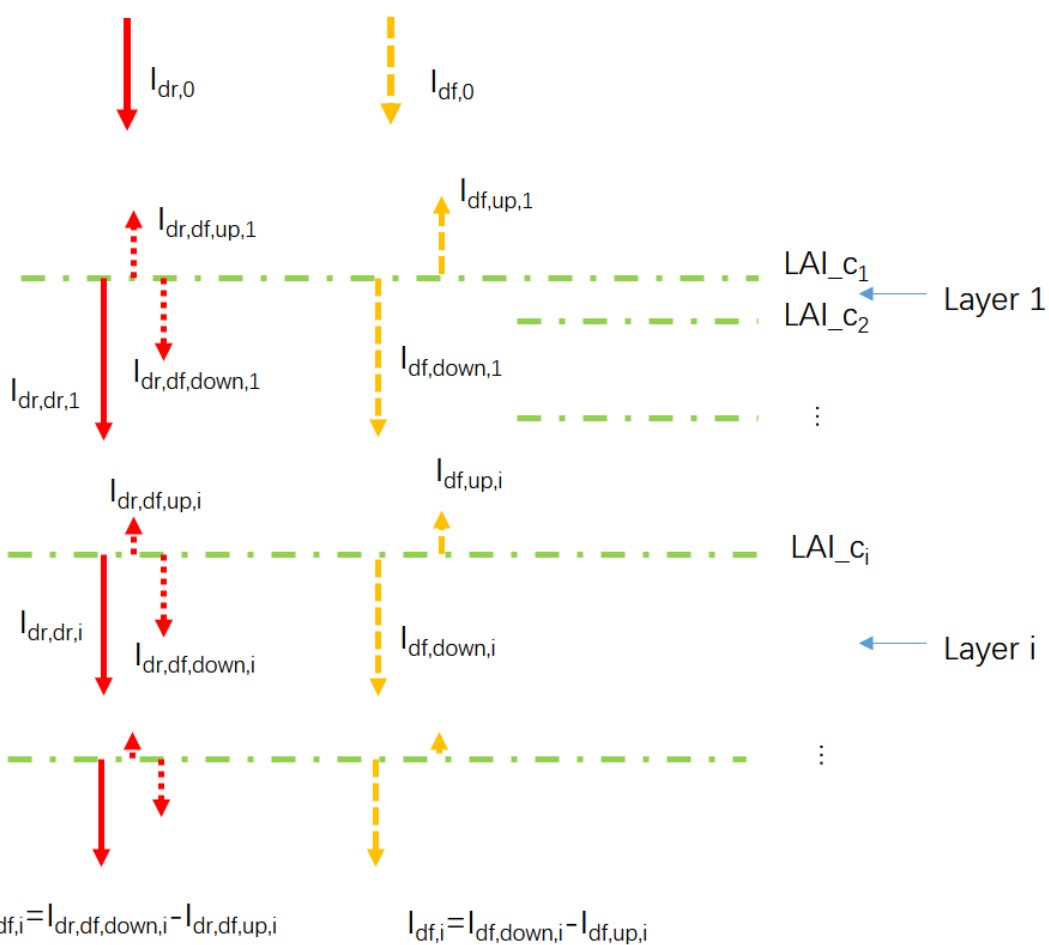

**Figure 3: The diagram of the canopy light transmission in ORCHIDEE_DF.** $I_{dr,0}$, downward direct PPFD at the top of the canopy; $I_{df,0}$, downward diffuse PPFD at the top of the canopy; $LAI\_c_i$, cumulative LAI above canopy layer $i$; $I_{dr,dr,i}$, downward direct PPFD at the top of canopy layer $i$; $I_{dr,df,i}$, net diffuse PPFD derived from the scattering of $I_{dr,0}$ at the top of canopy layer $i$, equals to the difference of its downward ($I_{dr,df,down,i}$) and upward ($I_{dr,df,up,i}$) components; $I_{df,i}$, net diffuse PPFD derived from $I_{df,0}$ at the top of canopy layer $i$, equals to the difference of its downward ($I_{df,down,i}$) and upward ($I_{df,up,i}$) components.





**Figure 4: Performance of ORCHIDEE trunk and ORCHIDEE_DF at different PFTs. (a) the Taylor plot of GPP, all valid 30min observations are used as reference, the filled circles indicate ORCHIDEE trunk, opened circles indicate ORCHIDEE_DF. (b) comparison of the correlation coefficients between the two models against observations. (c) and (e) same as (a) but for cloudy (diffuse light fraction >0.8) and sunny (diffuse light fraction<0.4) conditions only. (d) and (f) same as (b) but for cloudy and sunny conditions only.**

**Figure 5: Observed GPP and GPP modeled by ORCHIDEE trunk and ORCHIDEE_DF under cloudy (diffuse light fraction <0.4) conditions at selected sites (with relatively long time series) from each PFT.**



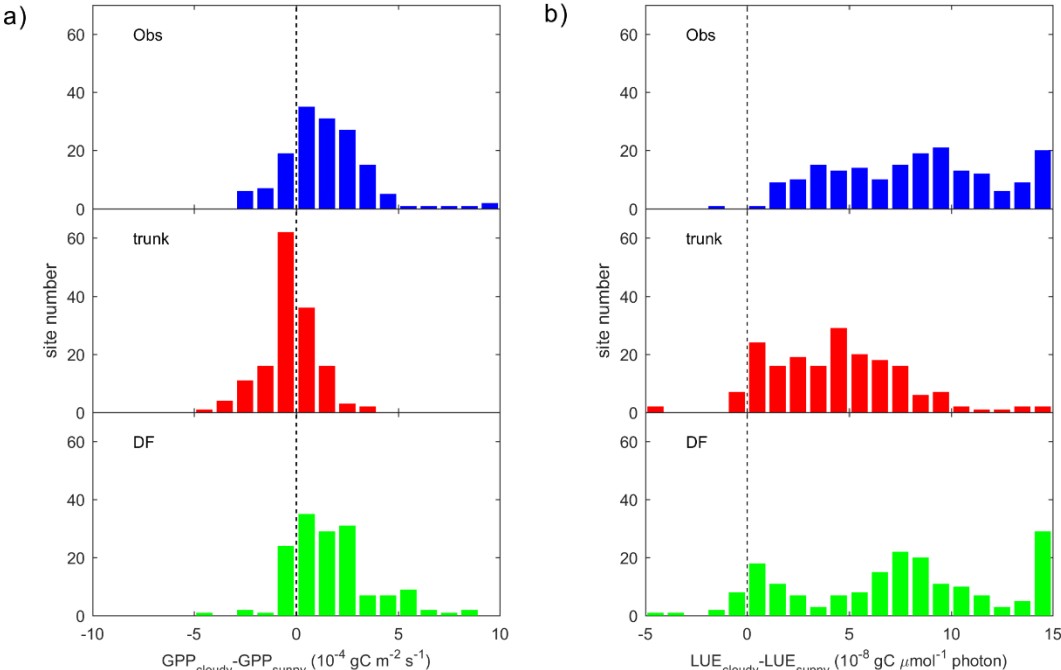

**Figure 6: Site distribution of (a) the GPP difference between cloudy (diffuse light fraction >0.8) and sunny (diffuse light fraction<0.4) conditions. (b) same as (a) but for LUE. It should be noted that the light level is controlled the same for sunny and cloudy conditions at each site.**



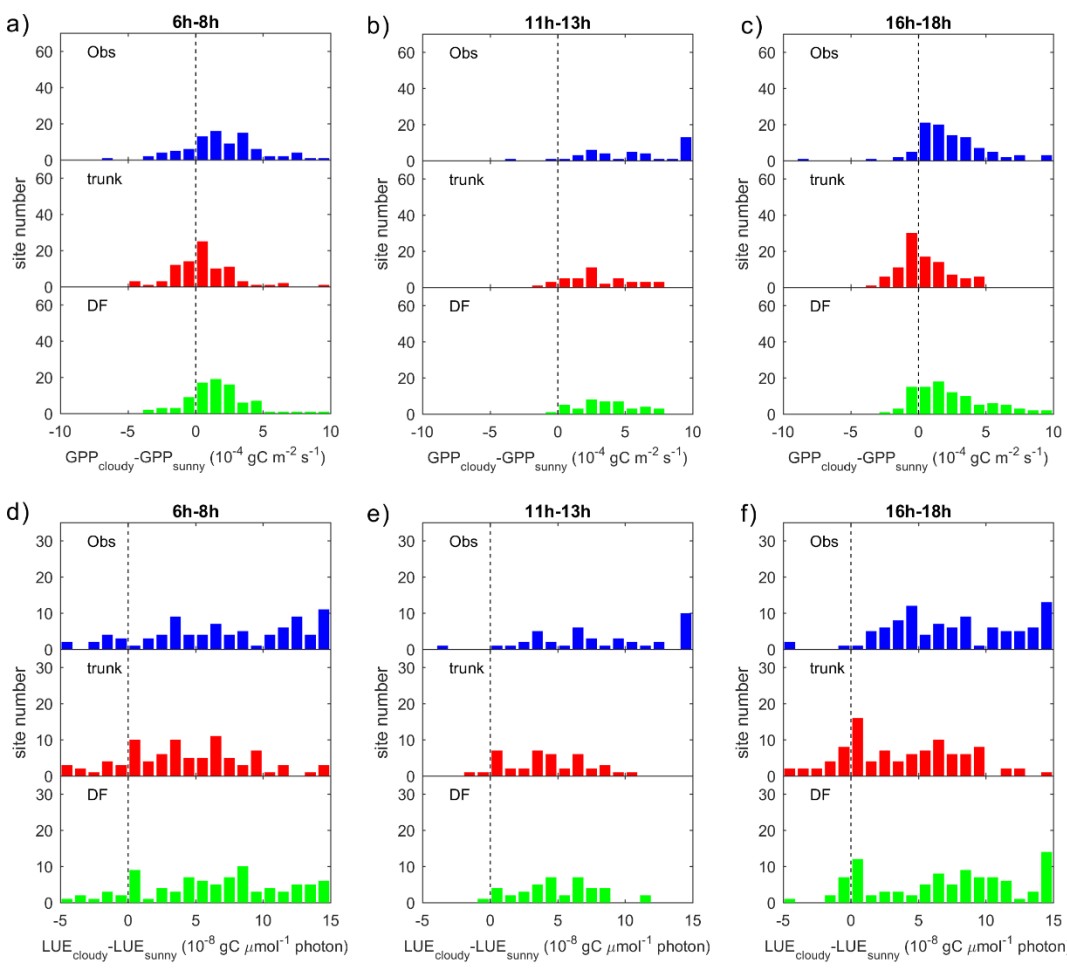

**Figure 7: Same as Figure 6 but differentiated for three times of the day.**





**Figure 8: The dependence of LUE difference between cloudy and sunny conditions on climate factors. In observation (blue), ORCHIDEE trunk (red) and ORCHIDEE_DF (green), the average and error bars indicate statistics of site level means (a) dependence of LUE difference on temperature, (b) dependence of LUE difference on VPD. (c) and (e) the same as (a) but for only forest sites and short vegetation (grasslands and croplands) sites. (d) and (f) the same as (b) but for forest sites and short vegetation sites.**





**Figure 9: The distribution of LUE difference between cloudy and sunny conditions (ΔLUE) in temperature-VPD field. The upper**
**numbers in each grid indicate the average of site level ΔLUE, while numbers in brackets indicate the number of sites with valid data.**
**(a) the ΔLUE based on observations, (b) the ΔLUE based on ORCHIDEE trunk, (c) the ΔLUE based on ORCHIDEE_DF**





Figure 10: Same as Figure 9 but for forests (a, c, e) and for short vegetation (b, d, f).

720





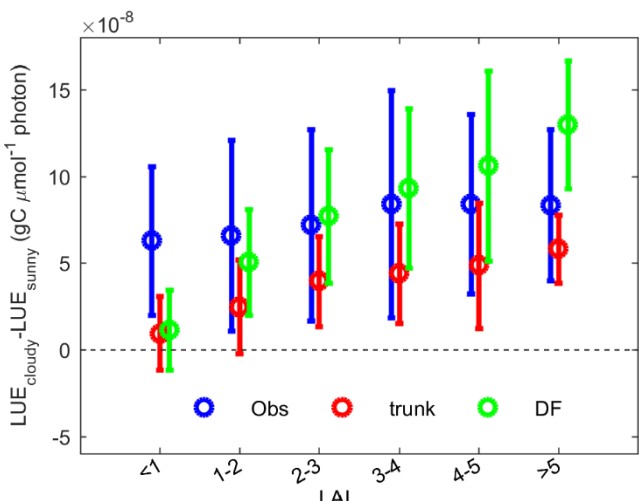

**Figure 11: Same as Figure 8a, but for LAI.**





**Table 1**. Variables in this study

| Variable | Definition | Unit |
|:---:|:---:|:---:|
| $A$ | Net photosynthesis rate | $\mu mol CO_2\ m^{-2}\ s^{-1}$ |
| $Ac$ | Rubisco activity limited net photosynthesis rate | $\mu mol CO_2\ m^{-2}\ s^{-1}$ |
| $Aj$ | Electron transport limited net photosynthesis rate | $\mu mol CO_2\ m^{-2}\ s^{-1}$ |
| $Cc$ | Chloroplast $CO_2$ partial pressure | $\mu bar$ |
| $Fdf_{PAR}$ | The fraction of diffuse PAR in total PAR | - |
| $Fdf_{PPFD}$ | The fraction of diffuse PPFD in total PPFD | - |
| $i$ | Leaf layer in canopy, for the top layer, $i=1$ | - |
| $I_0$ | Downward shortwave radiation at the top of the canopy | $W\ m^{-2}$ |
| $Iabs_{df,i}$ | Average absorption of $I_{df}$ per unit leaf area in canopy layer $i$ | $\mu mol\ m^{-2}s^{-1}$ |
| $Iabs_{dr,df,i}$ | Average absorption of $I_{dr,df}$ per unit leaf area in canopy layer $i$ | $\mu mol\ m^{-2}s^{-1}$ |
| $Iabs_{dr,dr,i}$ | Average absorption of $I_{dr,dr}$ per unit leaf area in canopy layer $i$ | $\mu mol\ m^{-2}s^{-1}$ |
| $Iabs_{dr,dr,i,sun}$ | Absorption of $I_{dr,dr}$ per sunlit unit leaf area in canopy layer $i$ | $\mu mol\ m^{-2}s^{-1}$ |
| $Iabs_{dr,i}$ | Average absorption of $I_{dr}$ per unit leaf area in canopy layer $i$ | $\mu mol\ m^{-2}s^{-1}$ |
| $Iabs_i$ | Average radiation absorption per unit leaf area in canopy layer $i$ | $W\ m^{-2}$ |
| $Iabs_{shd,i}$ | PPFD absorbed by shaded leaves per unit leaf area in canopy layer $i$ | $\mu mol\ m^{-2}s^{-1}$ |
| $Iabs_{sun,i}$ | PPFD absorbed by sunlit leaves per unit leaf area in canopy layer $i$ | $\mu mol\ m^{-2}s^{-1}$ |
| $I_{df,0}$ | Diffuse downward PPFD at the top of the canopy | $\mu mol\ m^{-2}s^{-1}$ |
| $I_{df,i}$ | Net PPFD derived from $I_{df,0}$ at the top of canopy layer $i$ | $\mu mol\ m^{-2}s^{-1}$ |
| $I_{dr,0}$ | Direct downward PPFD at the top of the canopy | $\mu mol\ m^{-2}s^{-1}$ |
| $I_{dr,df,i}$ | Net diffuse PPFD derived from the scattering of $I_{dr,0}$ at the top of canopy layer $i$ | $\mu mol\ m^{-2}s^{-1}$ |
| $I_{dr,dr,i}$ | Downward direct PPFD at the top of canopy layer $i$ | $\mu mol\ m^{-2}s^{-1}$ |
| $I_{dr,i}$ | Net PPFD derived from $I_{dr,0}$ at the top of canopy layer $i$, the sum of $I_{dr,dr,i}$ and $I_{dr,df,i}$ | $\mu mol\ m^{-2}s^{-1}$ |
| $I_i$ | Downward shortwave radiation arriving at canopy layer $i$ | $W\ m^{-2}$ |
| $J$ | Rate of electron transport | $\mu mol\ e^-\ m^{-2}\ s^{-1}$ |
| $Jmax$ | Maximum value of $J$ under saturated light, depending on temperature | $\mu mol\ e^-\ m^{-2}\ s^{-1}$ |





| | | |
|---|---|---|
| $Jmax_0$ | $Jmax$ at the top of the canopy | µmol e$^-$ m$^{-2}$ s$^{-1}$ |
| $Jmax_i$ | $Jmax$ at the canopy layer $i$ | µmol e$^-$ m$^{-2}$ s$^{-1}$ |
| $k$ | Light extinction coefficient in ORCHIDEE trunk | - |
| $k_b$ | Light extinction coefficient when leaves are assumed black | - |
| $k_d$ | Light extinction coefficient for diffuse PPFD | - |
| $KmC$ | Michaelis constants for $CO_2$, depending on temperature | µbar |
| $KmO$ | Michaelis constants for $O_2$, depending on temperature | µbar |
| $LAI\_c_i$ | Cumulative LAI above canopy layer $i$ | m$^2$ m$^{-2}$ |
| $LAIf_{shd,i}$ | Fraction of shaded leaf area in total leaf area in canopy layer $i$ | - |
| $LAIf_{sun,i}$ | Fraction of sunlit leaf area in total leaf area in canopy layer $i$ | - |
| $m$ | Optical air mass | - |
| $NIR_p$ | Potential total downward near infrared radiation at the top of the canopy | W m$^{-2}$ |
| $NIR_{p,df}$ | Potential diffuse downward near infrared radiation at the top of the canopy | W m$^{-2}$ |
| $NIR_{p,dr}$ | Potential direct downward near infrared radiation at the top of the canopy | W m$^{-2}$ |
| $NIR_{TOA}$ | Downward near infrared radiation at the top of the atmosphere | W m$^{-2}$ |
| $O$ | Chloroplast $O_2$ partial pressure | µbar |
| $p$ | Air pressure near surface | Pa |
| $p0$ | Standard sea level air pressure | Pa |
| $PAR_p$ | Potential total downward photosynthetically active radiation at the top of the canopy | W m$^{-2}$ |
| $PAR_{p,df}$ | Potential diffuse downward photosynthetically active radiation at the top of the canopy | W m$^{-2}$ |
| $PAR_{p,dr}$ | Potential direct downward photosynthetically active radiation at the top of the canopy | W m$^{-2}$ |
| $PAR_{TOA}$ | Downward photosynthetically active radiation at the top of the atmosphere | W m$^{-2}$ |
| $PPFDabs_i$ | Average photosynthetic photon flux density absorption per unit leaf area in canopy layer i | µmol m$^{-2}$s$^{-1}$ |
| $PPFD_{df}$, $I_{df,0}$ | Diffuse downward photosynthetic photon flux density above canopy | µmol m$^{-2}$s$^{-1}$ |
| $PPFD_t$ | Total downward photosynthetic photon flux density above canopy | µmol m$^{-2}$s$^{-1}$ |
| $R$ | Ratio of actual to potential downward shortwave radiation at the top of the canopy | - |
| $Rd$ | Dark respiration | gC m$^{-2}$ s$^{-1}$ |





| | | |
|---|---|---|
| $SW_{obs}$ | Actual (observed) downward shortwave radiation at the top of the canopy | $W\ m^{-2}$ |
| $SW_{p}$ | Potential (under clearsky conditions without clouds and aerosols) downward shortwave radiation at the top of the canopy | $W\ m^{-2}$ |
| $Vcmax$ | Maximum rate of Rubisco activity-limited carboxylation, depending on temperature | $\mu molCO_2\ m^{-2}\ s^{-1}$ |
| $Vcmax_0$ | Vcmax at the top of the canopy | $\mu molCO_2\ m^{-2}\ s^{-1}$ |
| $Vcmax_i$ | Vcmax at the canopy layer i | $\mu molCO_2\ m^{-2}\ s^{-1}$ |
| $\beta_{df}$ | Quanta-to-energy ratio for diffuse PAR | - |
| $\beta_t$ | Quanta-to-energy ratio for total PAR | - |
| $\Gamma^*$ | $CO_2$ compensation point in the absence of Rd | $\mu bar$ |
| $\theta$ | Solar zenith angle | degree |
| $\rho$ | The reflection coefficient of the canopy, i.e. the ratio between the downward and upward radiation at the top of the canopy | - |
| $\omega$ | Term accounting for atmospheric water vapor absorption | $W\ m^{-2}$ |

725