# Peer review of "Modeling the impacts of diffuse light fraction on photosynthesis in ORCHIDEE (v5453) land surface model"

_Geoscientific Model Development, 2020_

## Referee Comment (RC1) · Anonymous Referee #1 · 22 Jul 2020

**Review of "Modeling the impacts of diffuse light fraction on photosynthesis in ORCHIDEE (v5453) land surface model" by Zhang et al.**

**Summary**

There is a plethora of observational evidences supporting that diffused light favors higher plant productivity thanks to radiation being more evenly distributed and accessible across the shaded part of the canopy. Yet, few Land Surface Models (LSMs) have a representation of light transmission within the canopy that explicitly accounts for the quality of light (direct vs diffuse) and its effect of primary productivity. Zhang et al. implemented such a capability in the ORCHIDEE LSM (ORCHIDEE_DF) and this paper provides the details of their modelling framework as well as a solid evaluation of ORCHIDEE_DF performance against a large set of ground site measurements (FLUXNET).

This paper reads very easily as it is well written and well structured.

The first part of the paper describes the model framework and the parameterisations. It is generally well written, although some symbols / equations could be improved, and a couple of sections swapped together (see general comments).

The second part of the paper presents a very good evaluation of ORCHIDEE_DF. The added value of introducing a representation of diffuse light fraction is convincingly exposed and rigorous efforts were made to disentangle this from other cofounding effects (e.g .VDP, Temperature). Despite the calibration of model parameter being sub-optimal for this new configuration of ORCHIDEE, the analysis and supporting plots are very useful and effectively achieve to highlight where the model performs well, and which future development efforts should be prioritized. This is a useful evaluation effort which will also benefit the wider LSM community beyond the ORCHIDEE user base. The effort that the authors went through in evaluating ORCHIDEE_DF against a large ensemble of observations goes much further than previous attempts published in the literature and is greatly appreciated.

Adding a representation of diffuse light fraction in the canopy can only be useful if the boundary conditions – that is the fraction of diffuse radiation hitting the top of the canopy – is known. This information is usually lacking from the dataset that are used to drive LSMs. Technically, this is a problem that is external to land surface modelling, but it is great to see that Zhang et al. provide a practical framework to retrieve that missing information and could offer some insight to the terrestrial carbon cycle community for a developing a harmonized framework in future LSMs inter-comparisons.

The topic covered in this paper is absolutely relevant to GMD and I therefore strongly support its publication after addressing those minor very few points.

**General Comments**

1. I believe it will be improved at production stage, but some equations are not easy to read in current form. Use of upperscript and lowerscript could help bringing better separation between the terms in the equations (e.g. $Km_C$ instead of KmC, $C_c$ instead of Cc, etc).

2. Would it make more sense to introduce section 2.1.3 (Light transmission in ORCHIDEE_DF) before section 2.1.2 (Light partitioning in ORCHIDEE_DF) so it follows naturally section 2.1.1 (Light transmission in ORCHIDEE_trunk), especially given that the calculation of the fraction of diffuse light hitting the canopy top could eventually be treated by the radiative transfer of the driving

atmospheric model as it is done in an Earth System Framework (e.g. Yue et al., 2017; Malavelle et al. 2019) making a specific parameterization for this not necessary in ORCHIDEE?

3. For the evaluation framework described at P11 L324 to 326 – Getting the same level of PPFD that way may involves comparing GPPs at different time during the day which might not capture vegetation in similar physiological states. Wouldn't it be easier to simply normalize the cloudy and sunny GPPs by their respective PPFDs rather than removing a part of the dataset (likely the mid-day data for the sunny GPP when insolation is maximal and light saturation of the sunlit leaves possible)?

4. P13-L387-389 – It is interesting to note that both ORCHIDEE trunk and DF underestimate the dGPP and the dLUE around mid-day. Could it be related to the relative high proportion of sunlit leaves which is primarily a function of the solar zenith angle in the DF configuration? Segregating the dataset into latitudes may help to appreciate if this behaviour occurs more in the tropic or the mid-latitude sites.

**Specific Comments**

P02-L050 - VPD acronym has not been defined yet.
SC2: P02-L55 - "large-scale aerosol changes". [optional] You could add "and long-term changes in cloudiness".
P03-L075 - How come? Is it because of the large reduction in radiation under cloudy sky that tends to outweigh beneficial the effect of increased diffuse light?
P03-L076 - My bad, explanations for my comment above are provided in the following sentences. I would remove the word "Finally" which creates confusion during the transition between the two sentences.
P03-L077 - Williams et al. 2016 (year not matching the reference at the end, i.e. 2014).
P03-L087 - Le Quere et al. 2018 missing from the reference list.
P05-L143 - You can maybe point the reader towards Fig 3 as well. This schematic is useful for visualizing what eq. 4 calculates. I initially misunderstood what the cumulative LAI represente. It only represents the cumulative LAI above the current layer but does not include the current layer (if I got it right).
P05-L152 - Shouldn't it be $dI_i/dLAI\_c_i$ instead of $dI/dLAI\_c$ in eq. 6? What does the vertical bar symbol $|$ represents? Is it a derivative at fixed $LAI\_c_i$?
P06-L170 - Either explicitly provide the relationships or give a reference where those are documented.
P06-L175 - "forcing datasets": Do you mean dataset used to drive LSMs?
P06-L179 – $Fdf_{PAR}$. Should it be rewritten $fPAR_{df}$ to be consistent with the notation in other equations?
P07-L216 – Same as above, $fPPFD_{df}$ instead of $Fdf PPFD$?
P08-09 – eq. 26, 28, 30. Should it be LAI_ci instead of LAI_ai in the exponentials?
P08-L253 – Ref to Hikosaka et al. (2016) missing from the reference list.
P09-L268-270 – Same as eq. 6. The notation for the derivative is not clear to me. Could you explain?
P09-L289 – Change to "from 252 sites in total".
P10-L292 – Good job getting the references for all the sites!
P10-L292 – "annual climate" sounds weird. Could be rephrased by saying, "(climatological) annual mean temperature span the range xx to yy while (climatological) annual mean precipitation … span the range". Same for Fig S2 legend.
P13-L411 – This (fig 9 & 10) is an extremely useful way of presenting the sensitivity of the two models.
P14-L422 – Could that result be related to similarities in parameter traits and optimum points (e.g. Vcmax) between PFTs used to represent temperate and tropical biomes?

Figure 2 – The subtle light gradient makes it hard to appreciate the density of points. Could you maybe add a Probability Density Function along the x (respectively y) axis to represent the distribution of modelled (respectively observed) fraction of PPFD?

Figure 6 – "is controlled the same" feels a bit clunky. Could be rephrased by just saying that the sunny and cloudy days are sampled at equal light levels.

**Mentioned References**

*Malavelle, F. F., Haywood, J. M., Mercado, L. M., Folberth, G. A., Bellouin, N., Sitch, S., and Artaxo, P.: Studying the impact of biomass burning aerosol radiative and climate effects on the Amazon rainforest productivity with an Earth system model, Atmos. Chem. Phys., 19, 1301–1326, https://doi.org/10.5194/acp-19-1301-2019, 2019.*

*Yue, X. and Unger, N.: Aerosol optical depth thresholds as a tool to assess diffuse radiation fertilization of the land carbon uptake in China, Atmos. Chem. Phys., 17, 1329–1342, https://doi.org/10.5194/acp-17-1329-2017, 2017.*

---

## Referee Comment (RC2) · Anonymous Referee #2 · 5 Aug 2020

This manuscript led by Zhang presented a study on improving the ORCHIDEE land surface model with specific consideration of the impacts of diffuse light fraction on vegetation photosynthesis, a well recognized phenomenon but poorly represented in the existing version of ORCHIDEE model. The new model, named after ORCHIDEE_DF, has included a scheme for partitioning light into direct and diffuse components, and separated the existing multi-layer canopy into sunlit and shaded leaves with a two-stream radiative transfer model folowing Spitters 1986. Then the authors used global fluxnet observations to evaluate the new model and found that the new model better simulates GPP under different illumination conditions. Examinations on the effects of diffuse light on GPP and light use efficiency and the interactions between diffuse light

and other environmental factors such as temperature and vapor pressure deficit were conducted. The new model is suggested to have great potential in investigating aerosol effect on global biogeochemical cycles.

Overall the manuscript is very well organized and written, and easy to read. The description of the model development is clear, and the evaluation strategy is comprehensive and convincing. The analyses sections provide insightful understanding of the interactions of diffuse light and environmental factors. I don't really have much to add, but here I provide some minor suggestions and hope they can help further improve the quality of the manuscript.

1. Line 42-43: "However, this effect remains poorly represented in current land surface models". This is not accurate, at least CLM (Oleson et al., 2013), JULES (Mercado et al., 2009), CoLM (Dai et al 2004), iTem (Chen et al., 2014), and YIBs (Strada et al., 2016) have included processes that account for the diffuse light effect.

Oleson, K., Lawrence, D. M., Bonan, G. B., Drewniak, B., Huang, M., Koven, C. D., . . . Yang, Z. -L. (2013). Technical description of version 4.5 of the Community Land Model (CLM) (No. NCAR/TN-503+STR). doi:10.5065/D6RR1W7M Mercado LM, Bellouin N, Sitch S, et al. Impact of changes in diffuse radiation on the global land carbon sink. Nature. 2009;458(7241):1014-1017. doi:10.1038/nature07949 Dai, Y., R. E. Dickinson, and Y. Wang, 2004: A Two-Big-Leaf Model for Canopy Temperature, Photosynthesis, and Stomatal Conductance. J. Climate, 17, 2281–2299 Min Chen & Qianlai Zhuang (2014) Evaluating aerosol direct radiative effects on global terrestrial ecosystem carbon dynamics from 2003 to 2010, Tellus B: Chemical and Physical Meteorology, 66:1, DOI: 10.3402/tellusb.v66.21808 Strada, S. and Unger, N.: Potential sensitivity of photosynthesis and isoprene emission to direct radiative effects of atmospheric aerosol pollution, Atmos. Chem. Phys., 16, 4213–4234, https://doi.org/10.5194/acp-16-4213-2016, 2016.

The first three have been introduced in the paragraph of Line 88-100, but latter two

were directly applied for examining aerosol impacts and should be discussed as well.

2. I would suggest the authors provide a table of acronyms in Section 2.1.2 and 2.1.3 as an appendix so that the readers are easier to follow the equations.

3. Section 4.2 discussed factors affecting response of GPP to diffuse light and the authors suggested that the lower temperature and VPD may be the main cause of the higher midday GPP under cloudier conditions. Does ORCHIDEE simulate leaf temperature at different canopy layers? If not, it is not very convincing to me, as the short-term air temperature and VPD variations are mainly determined by the meteorological system, rather than the radiation regime.

4. Section 4.3. I think another important limitation of the developed ORCHIDEE_DF model for examining aerosol impacts is that it does not consider the impacts of the changing radiation regime on leaf temperature. This might be a second-order effect, but could be potentially important as shown in Chen and Zhuang, 2014 Tellus B.

Anyways, this is an excellent study and I recommend publish it with addressing the above minor points.

---

## Author Comment (AC1) · 3 Sep 2020

**Response to Reviewer #1**

**General comment:**

*"There is a plethora of observational evidences supporting that diffuse d light favors higher plant productivity thanks to radiation being more evenly distributed and accessible across the shaded part of the canopy. Yet, few Land Surface Models ( have a representation of light transmission within the canopy that explicitly accounts for the quality of light (direct vs diffuse) and its effect of primary productivity. Zhang et al. implemented such a capability in the ORCHIDEE LSM (ORCHIDEE_DF) and this paper provides the details of their modelling framework as well as a solid evaluation of ORCHIDEE_DF performance against a large set of ground site measurements (FLUXNET).*

*This paper reads very easily as it is well written and well structured.*

*The first part of the paper describes the model framework and the parameterisations. It is generally well written, although some symbols equations could be improved, and a couple of sections swapped together (see general comments).*

*The second part of the paper presents a very good evaluation of ORCHIDEE_DF The added value of introducing a representation of diffuse light fraction is convincingly exposed and rigorous efforts were made to disentangle this from other cofounding effects (e.g. VDP, Temperature). Despite the calibration of model parameter being sub-optimal for this new configuration of ORCHIDEE, the analysis and supporting plots are very useful and effectively achieve to highlight where the model performs well, and which future development efforts should be prioritized. This is a useful evaluation effort which will also benefit the wider LSM community beyond the ORCHIDEE user base. The effort that the authors went through in evaluating ORCHIDEE_DF against a large ensemble of observations goes much further than previous attempts published in the literature and is greatly appreciated.*

*Adding a representation of diffuse light fraction in the canopy can only be useful if the boundary conditions that is the fraction of diffuse radiation hitting the top of the canopy is known. This information is usually lacking from the dataset that are used to drive LSMs. Technically this is a problem that is external to land surface modelling, but it is*

*great to see that Zhang et al. provide a practical framework to retrieve that missing information and could offer some insight to the terrestrial carbon cycle community for a developing a harmonized framework in future LSMs inter-comparisons.*

*The topic covered in this paper is absolutely relevant to GMD and I therefore strongly support its publication after addressing those minor very few points."*

**[Response]** We thank the reviewer for the careful review and helpful comments and suggestions, which helped us to significantly improve our manuscript. We have addressed all the suggestions and comments in our revision. Please find below the reviewer's comments, followed by our responses and relevant changes in the manuscript. We hope that the revised version addresses all the issues raised by the reviewer.

**Comments:**

*"1. I believe it will be improved at production stage, but some equations are not easy to read in current form. Use of upper script and lower script could help bringing better separation between the terms in the equations (e.g. $Km_C$ instead of $KmC$, $C_c$ instead of $Cc$, etc)."*

**[Response]** We thank the reviewer for this suggestion, the notations have been improved throughout the manuscript. ($Ac$ to $A_c$, $Aj$ to $A_j$, $Rd$ to $R_d$, $Cc$ to $C_c$, $KmC$ to $Km_C$, $KmO$ to $Km_O$)

*"2. Would it make more sense to introduce section 2.1.3 (Light transmission in ORCHIDEE_DF) before section 2.1.2 (Light partitioning in ORCHIDEE_DF) so it follows naturally section 2.1.1 (Light transmission in ORCHIDEE_trunk) especially given that the calculation of the fraction of diffuse light hitting the canopy top could eventually be treated by the radiative transfer of the driving atmospheric model as it is done in an Earth System Framework (e.g. Yue et a l., 2017; Malavelle et al. 2019) making a specific parameterization for this not necessary in ORCHIDEE."*

**[Response]** We agree with the reviewer that it is more reasonable to put the two light

transmission sections together. Sections 2.1.2 and 2.1.3 have been swapped in the updated manuscript.

*"3. For the evaluation framework described at P 11 L 324 to 326 - Getting the same level of PPFD that way may involves comparing GPPs at different time during the day which might not capture vegetation in similar physiological states. Wouldn't it be easier to simply normalize the cloudy and sunny GPPs by their respective PPFDs rather than removing a part of the dataset (likely the midday data for the sunny GPP when insolation is maximal and light saturation of the sunlit leaves possible)?"*

**[Response]** Thanks for this question. We considered carefully and tried to use the proposed normalization method (Fig R1). The results are similar to what we found controlling PPFD level in the manuscript. However, we did not use it in the manuscript due to some concerns. It is known that the light response curve is not linear. Therefore, the LUE (GPP/PPFD) should depend on PPFD level. Due to the nature of atmospheric

[Figure]

Figure R1. Site distribution of the observed and modeled LUE difference between cloudy and sunny conditions at midday during the peak growing season (monthly GPP>90% of the mean monthly GPP maximum).

light transmission, the cloudy PPFD should be smaller than the sunny PPFD for a given solar zenith angle. If the PPFD level is not controlled, it would become difficult to explain whether the difference in LUE is due to diffuse radiation fraction or to the PPFD level. Therefore, in the manuscript, we compared the GPP and LUE with PPFD controlled at different times of the day (Fig. 7), which, we think, has ensured the vegetation to have similar physiological states in each period.

*"4. P 13 L 387 389 It is interesting to note that both ORCHIDEE trunk and DF underestimate the dGPP and the dLUE around midday. Could it be related to the relative high proportion of sunlit leaves which is primarily a function of the solar zenith angle in the DF configuration? Segregating the dataset in to latitudes may help to appreciate if this behaviour occurs more in the tropic or the mid latitude sites"*

**[Response]** Thanks for this point. We had made an extra analysis to investigate the latitude dependence of the ratio between modeled dGPP vs observed dGPP at midday (Fig. R2, two outlier sites not shown on the plot). There are no sites having similar midday PPFD level under sunny and cloudy conditions in low latitudes. According to the remaining data, positive relationship between the dGPP ratios and latitudes is not significant for both trunk and DF simulations. From our perspective, the underestimation in midday dGPP could be a result of parameterizations of processes other than diffuse radiation in ORCHIDEE because both the trunk and DF configurations have this problem. This will be added to the manuscript (Lines 387). "The underestimation of midday $\Delta$GPP could be a result of error in current ORCHIDEE parameterizations". With better parametrization and/or calibration done in the future,

this midday underestimation could be corrected.

**Minor comment:**

*P02 L050 VPD acronym has not been defined yet.*

**[Response]** It is now defined.

*SC2: P02 L 55 "large scale aerosol changes". [optional] You could add "and long term changes in cloudiness".*

**[Response]** It is added accordingly to the manuscript.

*P03 L 075 How come? Is it because of the large reduction in radiation under cloudy*

[Figure]

Figure R2. The dependence of the ratio between modeled cloudy-sunny GPP difference and observed GPP difference at midday (11:00-13:00). Neither ratios show a significant positive relationship with latitude.

*sky that tends to outweigh beneficial the effect of increased diffuse light?*

**[Response]** The reduction in radiation under cloudy sky can change the radiation budget at land surface and cause a cooling effect. This effect may decrease the VPD and mitigate its stress on stomatal conductance and finally affect GPP. The cooling itself can also influence directly photosynthesis rates in the model. Therefore, in the manuscript we wrote: "The covariance of these environmental factors may also cause the GPP to increase under cloudier conditions, although not being a direct effect of diffuse light".

*P03 L 076 My bad, explanations for my comment above are provided in the following sentences I would remove the word "Finally", which creates confusion during the transition between the two sentences.*

**[Response]** The word "Finally" has been changed to "Lastly" to avoid confusion.

*P03 L 077 Williams et al. 2016 (year not matching the reference at the end i.e. 2014).*

**[Response]** The year has been correctly accordingly.

*P03 L 087 Le Quere et al. 2018 missing from the reference list.*

**[Response]** The reference has been added to the list.

*P05 L 143 You can maybe point the reader towards Fig 3 as well. This schematic is useful for visualizing what eq. 4 calculates. I initially misunderstood what the cumulative LAI represente. It only represents the cumulative LAI above the current layer but does not include the current layer (if I got it right).*

**[Response]** Fig 3 is cited here. And yes the cumulative LAI above the current layer but does not include the current layer.

*P05 L 152 Shouldn't it be $dI_i /dLAI\_c_i$ instead of $dI/dLAI\_c$ in eq. 6 What does the vertical bar | symbol represents? Is it a derivative at fixed $LAI\_c_i$?*

**[Response]** Here $dI/dLAI\_c$ indicates the derivative of light with respect to cumulative LAI from the top of the canopy. Since this equation is continuous and for all canopy position, no subscript $i$ is added here. To calculate $Iabs_i$ which is the absorption at layer $i$, the derivative is calculated at layer $i$, noted $|LAI\_c_i$. This calculation is based on the assumption that all canopy layers are thin enough to neglect the difference in light absorption within each canopy layer (explained after Eq. 6).

*P06 L 170 Either explicitly provide the relationships or give a reference where those are documented.*

**[Response]** The reference has been added to the manuscript.

*P06 L 175 "forcing datasets": Do you mean dataset used to drive LSMs?*

**[Response]** Yes, the manuscript has been clarified to use "datasets to drive LSMs".

*P 06 L 179 $Fdf_{PAR}$ Should it be rewritten $fPAR_{df}$ to be consistent with the notation in other equations?*

*P 07 L 216 Same as above, $fPPFD_{df}$ instead of $Fdf_{PPF}$*

**[Response]** In the manuscript, we use $Fdf_{PAR}$ or $Fdf_{PPFD}$ to distinguish the fraction of diffuse light from the radiation variables in W m$^{-2}$ or in µmol m$^{-2}$ s$^{-1}$ using subscript "$_{df}$" for diffuse light (see Table 1).

*P08-09 eq. 26, 28, 30. Should it be $LAI\_c_i$ instead of $LAI\_a_i$ in the exponentials?*

**[Response]** Thanks for finding this error, the equations have been corrected.

*P08 L 253 Ref to Hikosaka et al. (missing from the reference list)*

**[Response]** The reference has been added to the list.

*P09 L 2 68 27 0 Same as eq. 6. The notation for the derivative is not clear to me. Could you explain?*

**[Response]** Please see the response to the above comment.

*P09 L 289 Change to "from 252 sites in total".*

**[Response]** The manuscript has been changed accordingly.

*P10 L 292 Good job getting the references for all the sites!*

*P10 L 292 "annual climate" sounds weird. Could be rephrased by saying, "(climatological) annual mean temperature span the range xx to yy while (climatological) annual mean precipitation ... span the range" Same for Fig S2 legend.*

**[Response]** It has been rephrased as: "The annual mean temperature of the sites spans from -9 to 27ºC, while the annual precipitation spans from 67 to over 3000 mm yr$^{-1}$" in the text. Fig S2 legend has been changed accordingly.

*P13 L 411 This (fig 9 & 10) is an extremely useful way of presenting the sensitivity of the two models*

*P14 L 422 Could that result be related to similarities in parameter traits and optimum*

*points (e.g. Vcmax) between PFTs used to represent temperate and tropical biomes?.*

**[Response]** The ORCHIDEE model calculates the optimum points for Vcmax according to the growth temperature the vegetation is adapting to during the season. The range of the acclimation spans from 11 to 35°C in the current ORCHIDEE model. Therefore, the model should be capable of distinguishing temperate and tropical biomes. However, considering the limiting observation data for calibration (P16 L 499), it is possible that current parameters are not good enough to represent sufficiently well temperature acclimation.

*Figure 2 The subtle light gradient makes it hard to appreciate the density of points. Could you maybe add a Probability Density Function along the x (respectively y) axis to represent the distribution of modelled (respectively observed) fraction of PPFD.*

**[Response]** Figure 2 has been improved accordingly (also Fig R3).

*Figure 6 "is controlled the same" feels a bit clunky Could be rephrased by just saying that the sunny and cloudy days are sampled at equal light levels*

**[Response]** The caption has been modified accordingly.

[Figure]

Figure R3. Modeled and observed diffuse PPFD fraction. (a) Scatter plot with the dark area indicates high data density, while light area indicates low data density, (b) Density distribution of the observed diffuse PPFD fraction, (c) Density distribution of the modeled diffuse PPFD fraction.

---

## Author Comment (AC2) · 3 Sep 2020

**Response to Reviewer #2**

**General comment:**

*"This manuscript led by Zhang presented a study on improving the ORCHIDEE land surface model with specific consideration of the impacts of diffuse light fraction on vegetation photosynthesis, a well recognized phenomenon but poorly represented in the existing version of ORCHIDEE model. The new model, named after ORCHIDEE_DF, has included a scheme for partitioning light into direct and diffuse components, and separated the existing multi-layer canopy into sunlit and shaded leaves with a two-stream radiative transfer model folowing Spitters 1986. Then the authors used global fluxnet observations to evaluate the new model and found that the new model better simulates GPP under different illumination conditions. Examinations on the effects of diffuse light on GPP and light use efficiency and the interactions between diffuse light and other environmental factors such as temperature and vapor pressure deficit were conducted. The new model is suggested to have great potential in investigating aerosol effect on global biogeochemical cycles.*

*Overall the manuscript is very well organized and written, and easy to read. The description of the model development is clear, and the evaluation strategy is comprehensive and convincing. The analyses sections provide insightful understanding of the interactions of diffuse light and environmental factors. I don't really have much to add, but here I provide some minor suggestions and hope they can help further improve the quality of the manuscript."*

**[Response]** We thank the reviewer for the review and helpful comments and suggestions, which helped us to further improve our manuscript. We have addressed all the suggestions and comments in our revision. Please find below the reviewer's comments, followed by our responses and relevant changes in the manuscript. We hope that the revised version addresses all the issues and satisfies the reviewer.

**Comments:**

*1. Line 42-43: "However, this effect remains poorly represented in current land*

*surface models". This is not accurate, at least CLM (Oleson et al., 2013), JULES (Mercado et al., 2009), CoLM (Dai et al 2004), iTem (Chen et al., 2014), and YIBs (Strada et al., 2016) have included processes that account for the diffuse light effect.*

*Oleson, K., Lawrence, D. M., Bonan, G. B., Drewniak, B., Huang, M., Koven, C. D., ... Yang, Z. -L. (2013). Technical description of version 4.5 of the Community Land Model (CLM) (No. NCAR/TN-503+STR). doi:10.5065/D6RR1W7M Mercado LM, Bellouin N, Sitch S, et al. Impact of changes in diffuse radiation on the global land carbon sink. Nature. 2009;458(7241):1014-1017. doi:10.1038/nature07949 Dai, Y., R. E. Dickinson, and Y. Wang, 2004: A Two-Big-Leaf Model for Canopy Temperature, Photosynthesis, and Stomatal Conductance. J. Climate, 17, 2281–2299 Min Chen & Qianlai Zhuang (2014) Evaluating aerosol direct radiative effects on global terrestrial ecosystem carbon dynamics from 2003 to 2010, Tellus B: Chemical and Physical Meteorology, 66:1, DOI: 10.3402/tellusb.v66.21808 Strada, S. and Unger, N.: Potential sensitivity of photosynthesis and isoprene emission to direct radiative effects of atmospheric aerosol pollution, Atmos. Chem. Phys., 16, 4213–4234, https://doi.org/10.5194/acp-16-4213-2016, 2016.*

*The first three have been introduced in the paragraph of Line 88-100, but latter two were directly applied for examining aerosol impacts and should be discussed as well.*
**[Response]** Thanks for the suggestion. We have added these studies to our updated manuscript: Line 42-43: "this effect remains poorly represented or evaluated in current land surface models." Line 92: "This two-big-leaf scheme was further used in iTem LSM (Chen and Zhuang, 2014) and got partly inherited in later CLM models (Oleson et al., 2013)." Line 100: "Apart from JULES, the Yale Interactive terrestrial Biosphere model (YIBs) also included a two-stream multilayer canopy light transmission scheme, but few efforts have been made to evaluate the ability of YIBs model to capture the observed diffuse light fertilization effect, especially at sub-daily time scales (Yue and Unger, 2015)."

*2. I would suggest the authors provide a table of acronyms in Section 2.1.2 and 2.1.3 as an appendix so that the readers are easier to follow the equations.*

[Response] An appendix section of acronym list has been added to the manuscript. After Line 524:

"**Appendix A**

List of acronyms:

Fdf:       Fraction of diffuse radiation

GPP:     Gross Primary Production

LAI:      Leaf Area Index

LSM:     Land Surface Model

LUE:      Light Use Efficiency

NIR:      Near-Infrared Radiation

PAR:     Photosynthetically Active Radiation

PFT:      Plant Functional Type

PPFD:   Photosynthetic Photon Flux Density

SW:       downward Shortwave Radiation at the top of canopy

TOA:     Top of Atmosphere

TOC:     Top of Canopy

VPD:     Vapor Pressure Deficit

* The variable names in Section 2 are listed in Table 1"

*3. Section 4.2 discussed factors affecting response of GPP to diffuse light and the authors suggested that the lower temperature and VPD may be the main cause of the higher midday GPP under cloudier conditions. Does ORCHIDEE simulate leaf temperature at different canopy layers? If not, it is not very convincing to me, as the short-term air temperature and VPD variations are mainly determined by the meteorological system, rather than the radiation regime.*

[Response] Thanks for raising this concern. In current ORCHIDEE trunk and DF model, the air temperature is taken directly as leaf temperature and does not vary

within the canopy. We agree that the short-term air temperature and VPD variations are mainly determined by the meteorological system. The explanation of leaf temperature is added to the manuscript: (Line 171) "Because in current ORCHIDEE, there is only one energy budget per grid cell, from which we cannot determine the leaf temperature, the air temperature is used to represent the leaf temperature in current model."

We compared the observed Tair and VPD under cloudy and sunny conditions at midday time and found that the cloudy midday Tair and VPD is lower than the sunny ones (Fig. R4). Therefore, the lower midday temperature and VPD could be the main cause of the detected midday ΔGPP in the manuscript. This lower cloudy midday Tair and VPD at site level might be because the time scales of weather systems which cause overcast conditions are often long enough to affect Tair. As a result, dynamics in canopy leaf temperature are not necessary to explain the simulated effect shown in the manuscript in line with the FLUXNET observations.

[Figure]

Figure R4. Site distribution of (a) the Tair difference (b) the VPD difference between cloudy (diffuse light fraction >0.8) and sunny (diffuse light fraction<0.4) conditions between 11:00 and 13:00.

*4. Section 4.3. I think another important limitation of the developed ORCHIDEE_DF model for examining aerosol impacts is that it does not consider the impacts of the changing radiation regime on leaf temperature. This might be a second-order effect, but could be potentially important as shown in Chen and Zhuang, 2014 Tellus B.*

**[Response]** Thanks for pointing out this limitation. Indeed, there remains no representation of the impacts of the changing radiation regime on leaf temperature in the current model, which may be potentially important. We have added some discussion of this point in Line 503 "Besides the possible bias in parameters, both ORCHIDEE trunk and ORCHIDEE_DF lack a representation of the response of leaf temperature to radiation. Instead, the air temperature is used directly to represent the leaf temperature throughout the canopy for simulating gas exchange processes in current model. As shown by Chen and Zhuang (2014), the changes of radiation regime due to aerosols can significantly affect leaf temperature, which could potentially affect GPP. For now, ORCHIDEE_DF remains not capable of dealing with this response of leaf temperature. Further developments are needed for disentangling the role of leaf temperature and diffuse light on GPP". This will be a future direction of our model development work.